# FedHERO: A Federated Learning Approach for Node Classification Task on Heterophilic Graphs

**Zihan Chen**                                                    *brf3rx@virginia.edu*
*Department of Electrical and Computer Engineering*
*University of Virginia*

**Xingbo Fu**                                                      *xf3av@virginia.edu*
*Department of Electrical and Computer Engineering*
*University of Virginia*

**Yushun Dong**                                                    *yd24f@fsu.edu*
*Department of Computer Science*
*Florida State University*

**Jundong Li**                                                  *jundong@virginia.edu*
*Department of Electrical and Computer Engineering*
*University of Virginia*

**Cong Shen**                                                    *cong@virginia.edu*
*Department of Electrical and Computer Engineering*
*University of Virginia*

## Abstract

Graph neural networks (GNNs) have shown significant success in modeling graph data, and Federated Graph Learning (FGL) empowers clients to collaboratively train GNNs in a distributed manner while preserving data privacy. However, FGL faces unique challenges when the general neighbor distribution pattern of nodes varies significantly across clients. Specifically, FGL methods usually require that the graph data owned by all clients is homophilic to ensure similar neighbor distribution patterns of nodes. Such an assumption ensures that the learned knowledge is consistent across the local models from all clients. Therefore, these local models can be properly aggregated as a global model without undermining the overall performance. Nevertheless, when the neighbor distribution patterns of nodes vary across different clients (e.g., when clients hold graphs with different levels of heterophily), their local models may gain different and even conflict knowledge from their node-level predictive tasks. Consequently, aggregating these local models usually leads to catastrophic performance deterioration on the global model. To address this challenge, we propose FED-HERO, an FGL framework designed to harness and share insights from heterophilic graphs effectively. At the heart of FEDHERO is a dual-channel GNN equipped with a structure learner, engineered to discern the structural knowledge encoded in the local graphs. With this specialized component, FEDHERO enables the local model for each client to identify and learn patterns that are universally applicable across graphs with different patterns of node neighbor distributions. FEDHERO not only enhances the performance of individual client models by leveraging both local and shared structural insights but also sets a new precedent in this field to effectively handle graph data with various node neighbor distribution patterns. We conduct extensive experiments to validate the superior performance of FEDHERO against existing alternatives.

# 1    Introduction

Graph Neural Network (GNN) aims to extract informative patterns from graphs (Xia et al., 2021; Wu et al., 2020b; Zhou et al., 2020; Scarselli et al., 2008; Xu et al., 2018a). However, graph data may contain sensitive information about the involved individuals (e.g., a user's race in a social network (Dwork et al., 2012; Beutel et al., 2017)). As a consequence, sharing graph data across different stakeholders is raising concerns due to the potential risk of privacy leakage, which further prevents training GNNs in a centralized manner (Cong & Mahdavi, 2022; Dai et al., 2022; Mei et al., 2019; Zhu & Zhu, 2020). To properly handle the above-mentioned problems, Federated Graph Learning (FGL) has emerged as a popular collaborative learning framework, which enables participants (clients) to derive insights from distributed graph sources while upholding stringent privacy safeguards (Liu et al., 2022a; Lou et al., 2021; Zhu et al., 2022; Zhang et al., 2022; Peng et al., 2021; Chen et al., 2021b). A typical scenario of FGL is that each client possesses a single graph (local graph) and seeks to collaboratively train a global GNN model for certain tasks (e.g., node classification tasks (Xiao et al., 2022; Bhagat et al., 2011)) without sharing the raw graph data. In such scenarios, current FGL approaches implicitly rely on the assumption of homophilic local graphs (Zhu et al., 2020; Ma et al., 2021), i.e., nodes within a client's graph are more likely to form connections with other nodes of the same class (Ma et al., 2019; Tang et al., 2009b). Specifically, when clients hold homophilic graphs, most of the neighbors of each node will have the same class as this center node (Ying et al., 2018). The GNNs trained on different clients will thus capture a similar tendency, and aggregating these GNNs usually makes the global model in FGL better learn such a tendency and benefit the generalization ability of the global model (Baek et al., 2023; Zhang et al., 2021).

Nevertheless, a critical issue regarding the above-mentioned assumption is that it overlooks the prevalence of heterophilic graphs (Pandit et al., 2007; Zheng et al., 2022; 2023; Bo et al., 2021; Yang et al., 2021; Chien et al., 2020; Liu et al., 2023a), where nodes belonging to different classes tend to be connected. For instance, in financial transaction networks, nodes denote bank clients, where node labels are their credit risk and edges represent the financial interactions between them. Such a network can be heterophilic, and we show an example in Figure 1. Specifically, a 'Moderate Risk' customer in Bank A may transact with multiple 'Low Risk' individuals, whereas in Bank B, the 'Moderate Risk' customer engages primarily with 'High Risk' counterparts. When training GNNs to predict customer risk, Bank A's GNN learns the connection between 'Moderate Risk' and 'Low Risk,' while Bank B's GNN captures the connection between 'Moderate Risk' and 'High Risk.' Directly aggregation models could lead to poor performance in predicting 'Moderate Risk' customers, as they fail to establish a consistent pattern that generalizes well across both banks. Consequently, this mismatch can result in performance degradation for FGL algorithms.

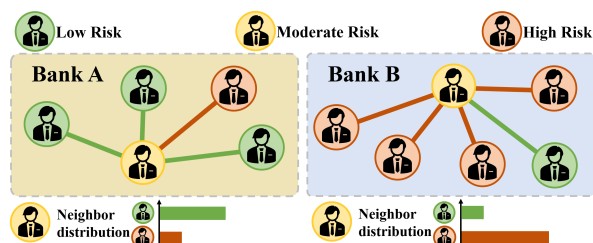

Figure 1: An example of financial transaction networks in two banks. The edges in the figure represent transaction records. Due to the diverse customer financial habits in distinct networks, customers (nodes) within the same risk classification (label) across different banks have diverse transactional relationship patterns (neighbor distribution).

To properly handle the challenge discussed above, in this paper, we introduce FEDHERO, a novel framework for **Fed**erated Learning on **He**te**ro**philic Graphs. FEDHERO is equipped with a carefully designed dual-channel GNN model, including the global channel and the local channel. The dual-channel design aims to aggregate GNNs that have learned similar patterns to facilitate knowledge sharing. To ensure each client obtains a GNN model aligned with others, we introduce a shared structure learning model that generates latent graphs across clients based on common patterns. By learning from and sharing the knowledge from these latent graphs, GNNs in the global channel capture similar tendencies, enhancing their overall performance. Meanwhile, the local channel operates directly on the original graph. With such a design, FEDHERO facilitates the sharing of common knowledge across clients while leveraging the unique structural information in each client's local graph to enhance overall model performance. This approach offers two key

advantages: (1) The global channel, trained on the latent graph, reduces reliance on the original neighbor distribution. The GNN models trained on different clients could capture similar tendencies from the shared structure learning model, and aggregating these GNNs could exploit FGL's potential. (2) Retaining the locally trained GNN on the original graph preserves sensitive information, enhancing privacy protection. We summarize the contributions of our paper below.

- **Problem Formulation.** To the best knowledge, we take an initial step to formulate and handle the challenge arising from the implicit heterophilic assumption of FGL.

- **Algorithmic Design.** We introduce an innovative FGL framework, named FEDHERO, which empowers clients to acquire global structure learning models while preserving graph-specific structural insights at the local level.

- **Empirical Evaluation.** We perform comprehensive experiments across four real-world graph datasets and three synthetic datasets. The results demonstrate that FEDHERO consistently outperforms existing FGL alternatives significantly.

## 2 PRELIMINARIES

**Notations.** Let $\mathcal{G} = (\mathcal{V}, \mathcal{E})$ be a graph containing $N$ nodes $\mathcal{V}$ and $\mathcal{E} \subseteq \mathcal{V} \times \mathcal{V}$ is the edge set. We define $\mathbf{X} \in \mathbb{R}^{N \times d}$ as the feature matrix, where $\mathbf{x}_v \in \mathbb{R}^d$ represents the attributes of node $v$. The initial graph structure can be represented in an adjacency matrix $\mathbf{A} = \{a_{uv}\}_{N \times N}$, where $a_{uv} = 1$ indicates the presence of an edge between node $u$ and $v$ and 0 otherwise. $\mathbf{Y} \in \mathbb{R}^{N \times C}$ denotes the label matrix, where $\mathbf{y}_v$ represents the label vector for node $v$, and $C$ is the number of classes. GNNs operate by representing nodes based on information from both their neighborhoods and their own attributes, which can be expressed as follows:

$$\mathbf{z}_v^l = \text{UPD}(\mathbf{z}_v^{l-1}, \text{AGG}(\{\mathbf{z}_u^{l-1} : \forall u \text{ s.t. } a_{uv} = 1\})), \tag{1}$$

where $\mathbf{z}_v^l$ is the embedding of the node $v$ at $l$-th layer. The operation AGG aggregates the embeddings of $v$'s neighbors, while UPD updates the representation of node $v$ using its embedding from the previous layer and the aggregated embeddings from its neighbors. Initially, $\mathbf{z}_v^0$ is set as $\mathbf{x}_v$.

**Federated Learning.** The objective of clients in FL is to jointly train models using local data. Let's consider $M$ data owners, each possessing data exclusively accessible to them, denoted as $\mathcal{D}_i = (\mathcal{X}_i, \mathcal{Y}_i)$, where $\mathcal{X}_i$ represents the data instances and $\mathcal{Y}_i$ is the corresponding labels. $N_i$ signifies the number of data samples held by client $i$ and $N = \sum_i N_i$ denotes the total number of data samples available. $\mathcal{L}_i$ and $w_i$ are the loss function and model parameters of client $i$, respectively. The FL process typically encompasses the following steps: (i) At $r$-th round, the server selects a subset of clients that participate in the training process. These clients receive the global model from the last round $\overline{w}_r$ and initialize the local model parameters as $w_i^r \leftarrow \overline{w}_r$. (ii) Each selected client updates its local model using its own local data to minimize the task loss: $w_i^{r+1} \leftarrow w_i^r - \eta \nabla \mathcal{L}_i(w_i^r)$. (iii) The server aggregates the updated local models to obtain an updated global model: $\overline{w}_{r+1} = \sum_i \frac{N_i}{N} w_i^{r+1}$. This process continues iteratively until convergence is achieved.

**Preliminary Study.** To highlight the unique challenges posed by varying neighbor distribution patterns across clients, we conduct a preliminary study comparing the performance of local training models with the global model produced by FEDAVG. Table 1 contrasts results on heterophilic datasets (Rozemberczki et al., 2021) (Squirrel and Chameleon) with those on homophilic datasets (PubMed (Namata et al., 2012) and Citeseer (Sen et al., 2008)). On homophilic graphs, the global model generally shows better performance. In contrast, for heterophilic graphs, the aggregated global model underperforms relative to local training models. This discrepancy arises because clients with heterophilic graphs are more likely to exhibit diverse neighbor distribution patterns, leading locally trained GNNs to capture varied tendencies. As a result, the global model struggles to aggregate and generalize this knowledge, resulting in poor predictions. This preliminary experiment underscores the impact of distinct neighbor distribution patterns on FGL performance, reinforcing the need to address the challenges identified in our study.

Table 1: Performances of the locally trained model and FEDAVG's global model on four datasets.

| Datasets | Squirrel | Chameleon | PubMed | Citeseer |
|---|---|---|---|---|
| | heterophilic | | homophilic | |
| LOCAL | 32.92±1.94 | 47.45±0.49 | 84.47±0.47 | 67.89±0.54 |
| FEDAVG | 28.55±0.45 | 38.65±2.19 | 87.08±0.01 | 72.41±0.22 |

**Problem Setup.** In the FGL system, we assume that $M$ clients hold $M$ graphs and aim to collaboratively train node classification models. In particular, client $i$ owns the graph $\mathcal{G}_i = (\mathcal{V}_i, \mathcal{E}_i)$, and $\mathbf{A}_i$ denotes the corresponding adjacency matrix of graph $\mathcal{G}_i$.

We aim to train a universal structure learner $g_L$ parameterized by $\theta$ across local graphs. This learner should capture the universally applicable patterns across graphs $\{\mathcal{G}_1, \mathcal{G}_2, ..., \mathcal{G}_M\}$. The trained structure learner is expected to generate effective graph structures that enhance the downstream classification performance of local GNN classifiers. Formally, the goal of training $g_L$ alongside the local models $f_i$ can be expressed as an optimization problem:

$$\min_{\theta, w_1, ..., w_M} \sum_{i=1}^{M} \mathcal{L}_i(f_i(g_L(\mathbf{X}_i, \mathbf{A}_i), \mathbf{X}_i, \mathbf{A}_i), \mathbf{Y}_i), \tag{2}$$

where $g_L$ takes the node feature $\mathbf{X}_i$ and adjacency matrix $\mathbf{A}_i$ of the local graph as input to generate a new graph structure, and the local model $f_i$ predicts the node classes from the new model structure and the original graph data.

## 3 Proposed Framework: FedHERO

We aim to address the issue of distinct neighbor distribution patterns, particularly for heterophilic graphs. In this context, clients possess heterophilic graphs in which the neighbor distribution patterns of nodes vary across different clients. As discussed in Section 1, this variability can result in diverse aggregation effects. Models trained on graphs with distinct neighbor distributions exhibit limited generalization ability.

To address this challenge, we present FEDHERO, as illustrated in Figure 2. First, we outline the implementation of structure learning and its initialization. We then demonstrate how FEDHERO integrates the global graph structure pattern and local structural information, detailing the joint optimization process involving structure learning and personalized task models. Lastly, we explore the mechanism for sharing structural knowledge within FEDHERO.

### 3.1 Graph Structure Learning in FedHERO

Graph structure learning endeavors to comprehend the message-flow pattern throughout the graph (Xia et al., 2021). Nevertheless, in FL scenarios where data is diversely distributed and privacy concerns are paramount, directly aggregating all data to train a comprehensive structure learning model becomes unfeasible. To tackle this challenge, we implement a structure learning model for each client, which can learn structural knowledge from local data. By sharing the learned knowledge with other clients, the entire FGL system can acquire a unified, globally optimized structural representation that captures universally applicable patterns across diverse graphs.

The objective of the structure learning model is to generate relevant latent structures that facilitate effective message passing based on the original graph. A common approach considers representing the weight of an edge between two nodes as a distance measure between two end nodes (Zhao et al., 2021b; Kazi et al., 2022; Chen et al., 2020). We follow this idea and employ a one-layer GNN to derive node representations $\mathbf{Z}$ from the original graph. The structure learning model can be represented by learning a metric function $\phi(\cdot, \cdot)$ of a pair of representations: $\tilde{a}_{uv} = \phi(\mathbf{z}_u, \mathbf{z}_v)$, where $\mathbf{z}_u$ is the learned embedding of node $u$ produced by GNN. The metric function $\phi(\cdot, \cdot)$ can assume various forms like cosine distance (Nguyen & Bai, 2010) and RBF kernel (Li et al., 2018b). In this study, we employ the multi-head attention mechanism as the metric

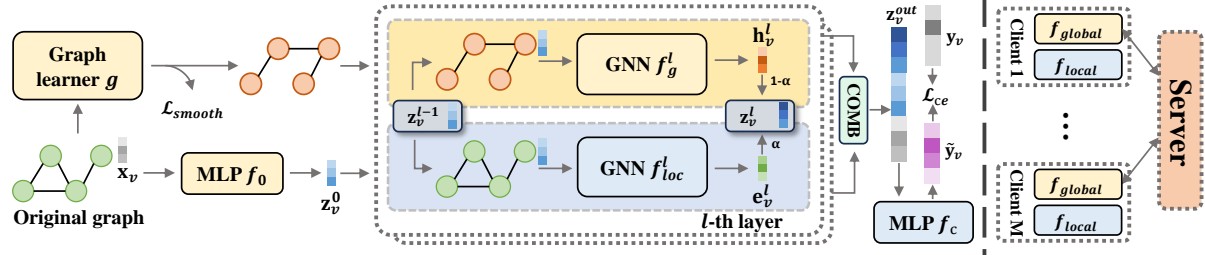

Figure 2: The framework of FEDHERO. Blue boxes represent models in the $f_{local}$ channel, which are trained locally and personalized for each client. Yellow boxes denote models in the $f_{global}$ channel, shared among clients to train a structure learner for mutual benefit. On the right side of the figure, the model aggregation scheme in FEDHERO is depicted.

function, following the approach outlined by Chen et al. (2020):

$$\phi(\mathbf{z}_u, \mathbf{z}_v) = \frac{1}{N_H} \sum_{h=1}^{N_H} \cos(\mathbf{w}_h^1 \odot \mathbf{z}_u, \mathbf{w}_h^2 \odot \mathbf{z}_v), \tag{3}$$

where $\cos(\cdot, \cdot)$ is cosine similarity, and $\odot$ denotes the Hadamard product. By consolidating the outputs $N_H$ heads and using $\mathbf{w}_h^1, \mathbf{w}_h^2$ to scale the node embeddings individually, the structure learning model can potentially identify connections between similar or dissimilar nodes and exhibit the necessary expressive power to handle graphs with heterophily. First, aggregating outputs of different heads can enhance the model's expressiveness in capturing the underlying influence between nodes from various perspectives. Additionally, the weight vectors $\mathbf{w}_h^1$ and $\mathbf{w}_h^2$ are introduced to learn element-wise scaling of input vectors, i.e., node representations. Moreover, the use of two independent weight vectors $\mathbf{w}_h^1$ and $\mathbf{w}_h^2$ allows for potential alignment or misalignment, enabling the model to connect both similar and dissimilar nodes. It's important to highlight that FEDHERO can be integrated with various metric functions and even various structure learning methods. Further empirical results are illustrated in Section 4.4.

To construct the latent graph $\tilde{\mathbf{A}} \in \mathbb{R}^{N \times N}$, we implement a post-processing step to create a $k$NN graph, wherein each node connects to up to $k$ neighbors, i.e.,

$$\tilde{\mathbf{A}}_{ij} = \begin{cases} \tilde{a}_{ij}, & \text{if } \tilde{a}_{ij} \text{ is a top-}k \text{ element in } \{\tilde{a}_{ij} | j \in [n]\}, \\ 0, & \text{otherwise.} \end{cases} \tag{4}$$

This design choice is based on the differentiability of the top-$k$ function, which facilitates the computation of parameter gradients and subsequent model updates in FEDHERO. Additionally, we explore an alternative method for latent graph generation. Such a method treats each edge in the latent graph as a Bernoulli random variable. Here, each latent edge $\tilde{\mathbf{A}}_{ij}$ is sampled from a Bernoulli distribution with parameter $\tilde{a}_{ij}$ (Wu et al., 2020a; Elinas et al., 2020; Zhao et al., 2023). A comparison of these methods and the impact of varying $k$ values is provided in Appendix C.

## 3.2  Optimization of the Dual-channel Model

In existing FGL frameworks (Zhang et al., 2021; Baek et al., 2023), the local models primarily employ one or multiple GNNs in sequence. These models initiate the process of generating node representations directly from raw features. However, as Tan et al. (2023) pointed out, directly constructing and sharing these local models across diverse clients can significantly compromise personalized performance due to feature and representation space misalignment. To address this challenge, divide the local model into two channels: $f_{global}$ and $f_{local}$. The global learner $f_{global}$ takes the original input graph and refines the latent graph structure. Through cooperation with other clients, $f_{global}$ detects patterns of universal relevance across heterophilic graphs. On the other hand, the local task learner $f_{local}$ extracts valuable client-specific information from

raw node features and structures within the local graph. This information includes biased structures or diverse feature distributions tailored for follow-up tasks.

In a client's local graph $\mathcal{G} = (\mathcal{V}, \mathcal{E})$, we initiate the process by utilizing a Multi-Layer Perception (MLP) $f_0$ to extract information from the node feature matrix $\mathbf{X}$ into node embeddings $\mathbf{Z}^0$. Next, we feed the node embeddings $\mathbf{Z}^0$, the latent graph $\tilde{\mathbf{A}}$, and the original graph structure $\mathbf{A}$ into an $L$-layer GNN. The middle part of Figure 2 depicts how FEDHERO leverages information from both the local graph structure $\mathbf{A}$ and the structure $\tilde{\mathbf{A}}$ from the structure learner to obtain node representations. In the $f_{global}$ channel (models in yellow boxes in Figure 2), the $l$-th layer of the GNN (denoted as $f_g^l$) utilizes the node embeddings from the previous layer $\mathbf{Z}^{l-1}$ and the latent graph $\tilde{\mathbf{A}}$ to generate hidden node embeddings that reveal patterns that hold universally across different graphs, denoted as $\mathbf{H}^l$. While feature propagation on the $\tilde{\mathbf{A}}$ could potentially yield enhanced node embeddings that align with global patterns, it is important to note that the original graph structures also contain valuable information, such as effective inductive bias (Bronstein et al., 2017). Therefore, we leverage the information within $\mathbf{A}$. It is combined with the output from the last layer, $\mathbf{Z}^{l-1}$, and fed into the $l$-th layer of the GNN (denoted as $f_{loc}^l$) within the $f_{local}$ channel (models in blue boxes in Figure 2). This process generates hidden node embeddings that adhere to the message-passing patterns within the local graph, resulting in $\mathbf{E}^l$. These embeddings play a crucial role in computing the layer-wise updates for node representations:

$$\mathbf{Z}^l = \sigma\bigg(\alpha \mathbf{E}^l + (1 - \alpha)\mathbf{H}^l\bigg), \tag{5}$$

where $\sigma$ is an activation function and $\alpha$ is a trading hyper-parameter that controls the concentration weight on input structures. Additionally, integrating the outputs from both the $f_{global}$ and $f_{local}$ channels can enhance training stability. This is achieved by mitigating the influence of substantial variations in latent structures encountered during the training process within the $f_{global}$ channel.

Recent studies show that merging intermediate node representations from multi-hops enables the GNN to integrate nodes of the same class (Zhu et al., 2020; Abu-El-Haija et al., 2019). In addition, encoding the features of the ego node separately from the aggregated neighbor representations enhances node embedding learning (Zhu et al., 2020; Suresh et al., 2021; Yan et al., 2022). Thus, we merge the intermediate node representations generated by each layer of the dual-channel GNNs with the ego feature $\mathbf{X}$: $\mathbf{Z}^{out} = \text{COMB}(\mathbf{X}, \mathbf{Z}^0, ..., \mathbf{Z}^L)$. The COMB function can be implemented in several ways, including max-pooling, concatenation, or LSTM-attention (Xu et al., 2018b). In our approach, we specifically use *concatenation* for the COMB function. This choice is based on the observation that concatenation consistently outperforms other methods in similar contexts (Xu et al., 2018b; Zhu et al., 2020).

In the classification stage, the prediction of the node class is based on its final embedding:

$$\tilde{\mathbf{Y}} = \text{Softmax}(f_c(\mathbf{Z}^{out})), \tag{6}$$

where the classify layer $f_c$ gives the $C$ dimension prediction.

Alongside the classification loss, we incorporate a regularization term to constrain $\tilde{\mathbf{A}}$ :

$$\mathcal{L}_{smooth} = \lambda \sum_{u,v} \tilde{\mathbf{A}}_{uv} ||\mathbf{x}_u - \mathbf{x}_v||_2^2 + \mu ||\tilde{\mathbf{A}}||_F^2, \tag{7}$$

where $||\cdot||_F$ is the Frobenius norm. The first term in Eq. (7) evaluates the smoothness of the latent graph, operating under the assumption that nodes with smoother features are more likely to be connected. The second term in Eq. (7) serves the purpose of preventing excessively large node degrees. The hyperparameters $\lambda$ and $\mu$ govern the magnitude of the regularization effects. Hence, in conjunction with the cross-entropy loss $\mathcal{L}_{ce}$, we fine-tune the local model using the following loss function:

$$\mathcal{L}_{total} = \mathcal{L}_{ce}(\tilde{\mathbf{Y}}, \mathbf{Y}) + \mathcal{L}_{smooth}. \tag{8}$$

Minimizing the loss function in Eq. (8), the generated latent structure is leveraged to generate node embeddings, providing supplementary information for representation learning.

## 3.3 Model Aggregation in FedHERO

The right side of Figure 2 illustrates the process of clients sharing local models and receiving updated models from the server. The model aggregation process in FEDHERO resembles that of FEDAVG (McMahan et al., 2017). However, in the FEDHERO framework, the server **only** receives and aggregates models from the $f_{global}$ channel of clients instead of the whole model. This sharing mechanism offers two key advantages for FEDHERO. First, the structure learning model in the global channel generates latent graphs from node embeddings with relatively low dependence on the original graph structure. By sharing this model among clients, the global channel GNNs learn similar patterns from the generated latent graphs. Aggregating these aligned models enhances the effectiveness of FGL, leading to improved overall performance. Second, retaining the model $f_{local}$—which focuses on learning the message passing pattern within the local graph—as private and training it locally allows the local model to capture client-specific information, ultimately contributing to subsequent tasks. Let $w_g^i$ denote the parameters of the $f_{global}$ in client $i$. The server does the aggregation: $\overline{w}_g = \sum_i \frac{N_i}{N} w_g^i$ where $N_i$ represents the number of nodes in the client $i$'s graph, and $N$ denotes the total number of nodes in the graph data of all clients. Subsequently, the server transmits $\overline{w}_g$ to the clients. The clients then update model in the $f_{global}$ channel with $\overline{w}_g$ and commence local training for the ensuing round.

In FEDHERO, the information in the $f_{local}$ channel, including the final classifier layer $f_c$, is not shared. This separation distinguishes between global and personalized information within the local graph. In Section 4.5, we evaluated the impact of sharing different parts of the model. This empirical examination underscored the efficacy of maintaining the $f_{local}$ channel in private, further emphasizing the effectiveness of bifurcating the GNN into two distinct channels.

## 3.4 Computational and Communication Overhead

We analyze the additional computational and communication overhead introduced by the dual-channel design in FEDHERO and compare with current FGL methods, providing insights into its scalability and efficiency in various FL environments.

Let $M$ denote the total number of clients, $E$ represent the number of local training epochs, and $L$ signify the number of layers in the model architecture. For simplicity, let $d$ denote the larger dimension between the node feature dimension $d_x$ and the embedding feature dimension $d_{hidden}$. $n_{link}, \theta^d, d_{dg}$, and $d_{rw}$ regard method-specific parameters. In Table 2, we compute the additional overhead caused by the FGL algorithms compared to FEDAVG (McMahan et al., 2017). Regarding communication overhead, since a relatively small $N_H$ (e.g. $N_H = 4$) enables the structure learner to perform effectively, the additional cost is comparable with other FGL methods like FEDLIT. As for computational overhead, the primary increase in computational load arises from dual-channel GNN. From Table 2, we can observe that the computation cost of FEDHERO aligns with the FGL methods that scale **linearly** with the size of node and edge sets. Therefore, FEDHERO's design allows it to handle large-scale datasets effectively without introducing much computation burden. In Appendix E, we further report the practical metrics including running time and memory usage, showing that FEDHERO scales well on large datasets and is comparable with FGL baselines.

Table 2: Additional computational and communication over- head of FGL methods compared with FEDAVG, calculated per client per round.

| Methods | Computation cost | Communication overhead |
|---------|------------------|------------------------|
| GCFL (Xie et al., 2021) | $\mathcal{O}((M(Ld^2)^2 + MlogM)/E)$ | $\mathcal{O}(1)$ |
| FEDLIT (Xie et al., 2023) | $\mathcal{O}(2|\mathcal{E}|n_{link}dI + |\mathcal{V}|d^2 + 2n_{link}^2dI/E)$ | $\mathcal{O}((n_{link} - 1)Ld^2)$ |
| FEDPUB (Baek et al., 2023) | $\mathcal{O}(Ld^2 + Md/E)$ | $\mathcal{O}(1)$ |
| FEDSTAR (Tan et al., 2023) | $\mathcal{O}(2Ld(d|\mathcal{V}| + |\mathcal{E}|))$ | $\mathcal{O}(1)$ |
| FEDSAGE (Zhang et al., 2021) | $\mathcal{O}(|\mathcal{V}|(\theta^d + Ld^2))$ | $\mathcal{O}(|\mathcal{V}|d + Ld^2)$ |
| FEDHERO | $\mathcal{O}(Ld(d|\mathcal{V}| + |\mathcal{E}|) + N_Hd|\mathcal{V}|)$ | $\mathcal{O}(N_Hd + d^2)$ |

# 4 EXPERIMENTS

## 4.1 Experimental Settings

**Datasets.** We carry out experiments using well-known heterophily datasets such as Squirrel and Chameleon, derived from Wikipedia datasets (Rozemberczki et al., 2021), along with the Actor dataset (Tang et al., 2009a) and the Flickr dataset (Zeng et al., 2019). Given these datasets' inherent incompatibility with the FGL framework, we adapt them into a federated context using two prevalent graph partitioning methods: the METIS algorithm (Karypis & Kumar, 1997) and the Louvain algorithm (Blondel et al., 2008). This approach enables us to create *semi-synthetic* federated graph learning datasets. As detailed in Appendix A, the graph partitioning process preserves the heterophilic properties of the graphs. Additionally, Appendix B demonstrates significant differences in the neighbor distributions of same-class nodes across clients. This highlights the challenges in FGL that we discussed in the Introduction. We follow the common FGL protocol (Baek et al., 2023; Zhang et al., 2021) with non-overlapping clients for fair comparison with prior work. We note that FEDHERO does not rely on this assumption, since no component in FEDHERO relies on unique nodes among subgraphs, cross-client neighborhoods, or graph stitching. For further details on this process and the overlapping scenario, please refer to Appendix A and C, respectively.

We also include three graph datasets: syn-cora (Zhu et al., 2020), ieee-fraud (Howard et al., 2019), and credit (Yeh & Lien, 2009). These datasets naturally lend themselves to subdivision into multiple subgraphs (e.g., distinct subgraphs within the ieee-fraud dataset correspond to different product transaction records), and we denote them as *real-world* datasets. The description of the datasets is summarized in Appendix A.

**Hyper-Parameter Settings.** The architecture of the GNN models consists of a feature projection layer, two graph convolutional layers, and a classifier layer. We employ Adam optimizer (Kingma & Ba, 2014) with a learning rate of 0.005. We apply a consistent hyperparameter setting in our main results. Specifically, we set $\alpha = 0.2$, $\mu = \lambda = 0.1$, $k = 20$, and $N_H = 4$. In the supplementary material, we provide an analysis of hyperparameters, indicating that optimizing these values could lead to further improvements in performance. The training process involves 200 communication rounds, with each local training epoch lasting one iteration. The nodes are randomly distributed into five groups. For each experiment, one group is chosen as the test set, while three of the remaining groups are utilized for training and one for validation. We perform five experiments on each dataset and report the models' average performance. For the ieee-fraud and credit datasets, we present the mean test Area Under the Curve (AUC), accounting for the imbalanced labels. For the remaining datasets, we use mean test accuracy across clients for evaluation. All methodologies are implemented using PyTorch 1.8 and executed on NVIDIA RTX A6000 GPUs. Our code can be found in the supplementary material.

**Baselines.** We comprehensively compare FEDHERO against five baseline methods in federated graph learning. These baselines encompass strategies that focus on (i) promoting collaboration among clients with similar graph distributions (GCFL (Xie et al., 2021) and FEDPUB (Baek et al., 2023)), (ii) sharing structural insights (FEDSTAR (Tan et al., 2023)), (iii) producing missing edges between subgraphs (FEDSAGE (Zhang et al., 2021)), and (iv) discerning and modeling message-passing for latent link types (FEDLIT (Xie et al., 2023)). To ensure the completeness of our experiments, we also include the standard FL method FEDAVG (McMahan et al., 2017).

## 4.2 Evaluation on Semi-synthetic Datasets

We present the node classification results on four semi-synthetic heterophilic datasets. The average test accuracy and its standard deviation are summarized in Table 3. The results indicate that FEDHERO consistently outperforms all the baseline methods by a significant margin. This is because the global channel in FEDHERO has less dependence on the local graph structure, with the GNN $f_{global}$ being updated on the latent graph. By sharing the structure learning model across clients, FEDHERO ensures that the latent graphs generated by different clients exhibit similar neighbor distribution patterns. This allows the aggregation of consistent information from the global channel, ultimately enhancing the performance.

Table 3: Performance of FEDHERO and baselines on semi-synthetic heterophilic datasets. Bold fonts indicate the best performances among all methods, while underlines denote the next best results across all methods. M denotes the number of clients.

| Datasets | Squirrel | | | | Chameleon | | | | Actor | | | | Flickr | | | |
|---|---|---|---|---|---|---|---|---|---|---|---|---|---|---|---|---|
| | METIS | | | Louvain | METIS | | | Louvain | METIS | | | Louvain | METIS | | | Louvain |
| | M=5 | M=7 | M=9 | | M=3 | M=5 | M=7 | | M=5 | M=7 | M=9 | | M=30 | M=40 | M=50 | |
| LOCAL | 31.32 | 31.44 | 31.33 | 32.92 | 43.77 | 44.34 | 45.15 | 47.45 | 29.27 | 28.75 | 28.37 | 26.14 | 45.03 | 44.98 | 45.37 | 45.02 |
| | (±1.21) | (±0.11) | (±0.68) | (±1.94) | (±1.07) | (±3.16) | (±1.37) | (±0.49) | (±0.60) | (±0.50) | (±0.34) | (±0.90) | (±0.13) | (±0.36) | (±0.37) | (±0.26) |
| FEDAVG | 29.67 | 28.28 | 27.18 | 28.55 | 42.27 | 39.08 | 45.09 | 38.65 | 31.90 | 31.45 | 31.40 | 31.14 | 42.45 | 42.40 | 42.30 | 42.56 |
| | (±1.04) | (±0.64) | (±1.37) | (±0.45) | (±2.39) | (±1.15) | (±3.24) | (±2.19) | (±1.02) | (±0.72) | (±0.32) | (±1.58) | (±0.2) | (±0.16) | (±0.39) | (±0.19) |
| FEDPUB | 29.52 | 29.38 | 29.12 | 32.87 | 42.36 | 46.40 | 45.71 | 47.16 | 28.50 | 26.92 | 26.95 | 26.92 | 47.47 | 46.64 | 46.77 | 46.99 |
| | (±0.86) | (±1.04) | (±0.56) | (±1.94) | (±1.50) | (±1.29) | (±1.69) | (±2.93) | (±0.84) | (±0.99) | (±1.20) | (±1.15) | (±0.76) | (±1.28) | (±0.39) | (±0.85) |
| FEDSTAR | 31.74 | 32.60 | 33.53 | 36.84 | 50.60 | 50.42 | 50.72 | 51.32 | 28.58 | 27.13 | 27.18 | 27.43 | 49.35 | 49.15 | 49.08 | 49.03 |
| | (±0.73) | (±1.06) | (±0.91) | (±0.99) | (±0.84) | (±1.14) | (±1.19) | (±1.26) | (±0.74) | (±0.44) | (±0.40) | (±1.19) | (±0.18) | (±0.62) | (±0.25) | (±0.45) |
| GCFL | 31.40 | 30.04 | 28.98 | 30.42 | 45.12 | 43.47 | 40.30 | 38.73 | 31.39 | 31.63 | 31.41 | 31.63 | 47.65 | 47.41 | 46.88 | 47.37 |
| | (±2.09) | (±1.25) | (±0.13) | (±2.28) | (±1.44) | (±2.01) | (±2.81) | (±3.75) | (±0.94) | (±1.23) | (±1.23) | (±1.29) | (±0.22) | (±0.24) | (±0.50) | (±0.39) |
| FEDLIT | 30.01 | 28.22 | 27.42 | 32.76 | 40.10 | 34.36 | 35.97 | 36.67 | 32.23 | 31.74 | 32.16 | 32.85 | 48.18 | 48.35 | 48.56 | 48.49 |
| | (±2.46) | (±2.59) | (±2.42) | (±3.56) | (±4.45) | (±4.38) | (±3.80) | (±5.06) | (±1.07) | (±0.82) | (±1.13) | (±1.71) | (±0.59) | (±0.36) | (±0.44) | (±0.52) |
| FEDSAGE | 33.30 | 33.07 | 33.10 | 35.17 | 50.93 | 48.72 | 49.17 | 45.51 | 33.75 | 32.73 | 33.30 | 33.57 | 45.75 | 45.45 | 44.74 | 50.37 |
| | (±0.63) | (±0.88) | (±1.02) | (±2.13) | (±1.77) | (±2.61) | (±2.80) | (±1.86) | (±1.02) | (±0.87) | (±0.89) | (±1.83) | (±0.23) | (±0.28) | (±0.36) | (±0.56) |
| **FedHERO** | **36.63** | **36.08** | **37.83** | **38.00** | **53.93** | **54.06** | **52.65** | **55.63** | **35.25** | **34.60** | **34.32** | **34.54** | **50.51** | **50.33** | **49.96** | **50.45** |
| | (±0.95) | (±0.87) | (±1.02) | (±0.92) | (±0.88) | (±0.56) | (±1.33) | (±2.00) | (±0.29) | (±1.03) | (±0.68) | (±0.89) | (±0.32) | (±0.25) | (±0.31) | (±0.25) |

We also observed that FEDAVG often underperforms compared to local training in many experiments. This suggests that heterophilic graphs are more likely to exhibit distinct neighbor distribution patterns across them, which can negatively impact FEDAVG's performance. Notably, FEDPUB and GCFL demonstrate superior overall performance compared to FEDAVG. Both encourage clients to collaborate with those who have similar model gradients or outputs. During optimization, the GNN learns to capture graph information, including neighbor distribution patterns. Therefore, similar model gradients or outputs indicate similar neighbor distribution patterns or tendencies captured by the model. As previously discussed, aggregating models with similar tendencies enhances the performance of the aggregated model.

In our investigation of FEDHERO's performance with heterophilic data, we also observed significant improvements in datasets characterized by high heterophily, particularly in the Squirrel and Chameleon datasets. These datasets have homophily ratios of 0.22 and 0.25, respectively, as reported in (Mao et al., 2023), with lower ratios indicating greater heterophily. In comparison to Flickr (homophily ratio: 0.32 (Kim & Oh, 2022)), FEDHERO's enhancements are more pronounced. For example, under Louvain partitioning, FEDHERO demonstrates improvements of 4.31%, 1.16%, and 0.08% on the Chameleon, Squirrel, and Flickr datasets, respectively, when compared to the best-performing baseline. We believe this phenomenon occurs because, as heterophilic ratio increases, the differences in node neighbor distribution patterns across clients' local graphs become more pronounced, posing greater challenges to existing FGL methods.

## 4.3 Evaluation on Real-world Datasets

Figure 3 provides a comprehensive overview of the evaluation results across real-world datasets in FGL settings. FEDHERO consistently outperforms all FGL baselines on ieee-fraud and credit datasets. The similar performance between our method and other FGL baselines on ieee-fraud and credit datasets may be attributed to distinct class features that are easily distinguishable (e.g., a linear regression model can yield satisfactory results) (Howard et al., 2019) or the low dimensionality of features (Yeh & Lien, 2009). The superior performance of FEDSAGE on the syn-cora dataset

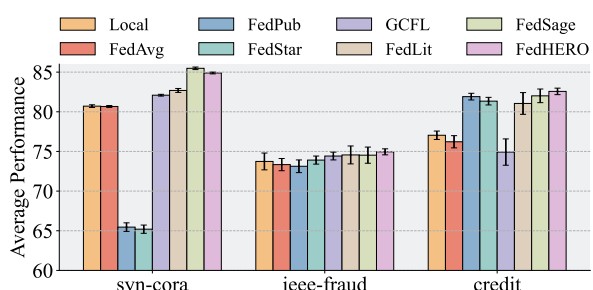

Figure 3: Performance of FEDHERO and baselines on three real-world datasets.

may be attributed to GraphSage's utilization of a hybrid design, incorporating elements of both ego-embedding and neighbor embedding. Similar findings are also observed in (Zhu et al., 2020).

Notably, FEDPUB and FEDSTAR face challenges in achieving strong performance on the syn-cora dataset. The syn-cora dataset consists of random graphs where nodes of different classes have the same expected degree, and different graphs share the same degree distribution (Abu-El-Haija et al., 2019; Zhu et al., 2020). The suboptimal performance of FEDSTAR also demonstrates that relying solely on structural information from the adjacency matrix, while overlooking distinctions between nodes connected by an edge, is insufficient for handling heterophilic graphs. Thus, structure embeddings in FEDSTAR don't contain distinguishable information across classes. Sharing the structure knowledge doesn't benefit training GNNs for node classification. On the other hand, FEDPUB assumes graphs in clients have community structures, and it leverages the community information. However, the assumption does not hold in syn-cora where random graphs are generated independently.

## 4.4 FedHERO with Different Structure Learning Methods

In this section, we explore the adaptability of FEDHERO to various structure learning techniques. For example, we substitute the similarity function with an attention-based structure learner, denoted as FEDHERO*att*, and with Graph Attention Networks (GAT) (Veličković et al., 2018), denoted as FEDHERO*gat*. Another variant, FEDHERO*cos*, uses cosine distance as the similarity function. We further examine FEDHERO with more complicated graph learning methods, including GLCN (Jiang et al., 2019), GAug (Zhao et al., 2021b). We replace the structure learning model in FEDHERO with these methods. The experiments are performed on the Squirrel and Chameleon datasets, and the results are presented in Table 4. From the results, we observe the following:

- The original FEDHERO, integrated with multi-head weighted attention, consistently outperforms its variants. This superior performance can be attributed to the ability of multi-head weighted attention to capture a wide range of influences between nodes effectively, making it particularly effective for heterophilic graphs.

- Among basic variants, FEDHERO*cos* shows slightly better performance than the other two variants, likely because it avoids additional model parameters, thus reducing the risk of overfitting. Additionally, unlike FEDHERO*gat*, FEDHERO*cos* considers the entire subgraph for edge generation, which mitigates the impact of heterophilic graphs on the GNN aggregation mechanism (Li et al., 2022; Yang et al., 2022).

Table 4: Performance of FEDHERO with different graph structure learning methods.

| Variants | Squirrel | | Chameleon | |
|---|---|---|---|---|
| | METIS M=9 | Louvain | METIS M=7 | Louvain |
| FEDHERO$_{cos}$ | 34.65±0.69 | 37.83±1.78 | 51.36±2.72 | 52.39±1.74 |
| FEDHERO$_{att}$ | 34.92±1.00 | 36.80±1.49 | 51.71±1.88 | 54.28±1.62 |
| FEDHERO$_{gat}$ | 34.83±0.81 | 37.12±1.16 | 51.45±1.73 | 53.26±2.43 |
| FEDHERO+GLCN | 35.11±1.14 | 37.25±0.85 | 52.18±3.63 | 52.54±2.29 |
| FEDHERO+GAug | 36.65±1.53 | 37.63±1.00 | 52.44±2.67 | 54.57±2.42 |
| FEDHERO | **37.83±1.02** | **38.00±0.92** | **52.65±1.33** | **55.63±2.00** |

## 4.5 FedHERO with Various Sharing Mechanisms

We conduct an analysis of various FEDHERO variants involving the sharing of different components of the local model, including all model parameters, task model parameters, and none (denoted as FEDHERO$_{all}$, FEDHERO$_{gcn}$, and FEDHERO$_{none}$, respectively). Additionally, we consider whether to incorporate a structure-task model decoupled scheme. For variants without the Dual-Channel (DC), we remove the GNN in the $f_{local}$ channel and feed the learned adjacency matrix and the original matrix into the same GNN.

Table 5: Performance of FEDHERO when sharing different modules with server.

| Variants | DC | Squirrel | | Chameleon | |
|---|---|---|---|---|---|
| | | METIS M=7 | Louvain | METIS M=7 | Louvain |
| FEDHERO$_{none}$ | - | 32.53±0.66 | 36.85±0.85 | 48.48±1.12 | 48.30±1.50 |
| FEDHERO$_{all}$ | - | 29.37±1.18 | 32.84±0.92 | 44.09±2.84 | 44.12±1.98 |
| FEDHERO$_{none}$ | ✓ | 34.14±1.35 | 36.25±2.96 | 50.62±1.54 | 52.03±1.70 |
| FEDHERO$_{all}$ | ✓ | 32.04±1.57 | 34.08±2.14 | 45.38±1.64 | 46.39±3.03 |
| FEDHERO$_{gcn}$ | ✓ | 28.05±0.65 | 31.13±1.57 | 42.27±1.67 | 43.70±1.86 |
| FEDHERO | ✓ | **37.83±1.02** | **38.00±0.92** | **52.65±1.33** | **55.63±2.00** |

In Table 5, we observe that the structure-task model decoupling scheme consistently outperforms the model without it, indicating that independently learning the structural knowledge enhances FEDHERO's performance across various scenarios. Furthermore, sharing all parameters or the task model results in performance degradation compared to pure local training. This observation underscores that sharing local task model information across heterophily graphs harms FEDHERO's performance, since the original graphs have different node neighbor distribution patterns and aggregating of models that capture distinct tendencies would lead to performance degradation for the global model.

## 4.6 Convergence Analysis

We conduct experiments to compare the convergence speed of FEDHERO with other FGL baselines on different datasets. The results, as presented in Figure 4, demonstrate that FEDHERO exhibits faster convergence compared to most baselines on both datasets. Notably, FEDHERO's convergence speed is comparable to that of FEDSTAR, since both two methods employ a dual-channel GNN in the client's model. We conclude that FEDHERO does not require more steps to converge compared to existing methods, though it introduces additional parameters, which underscores the efficiency of FEDHERO.

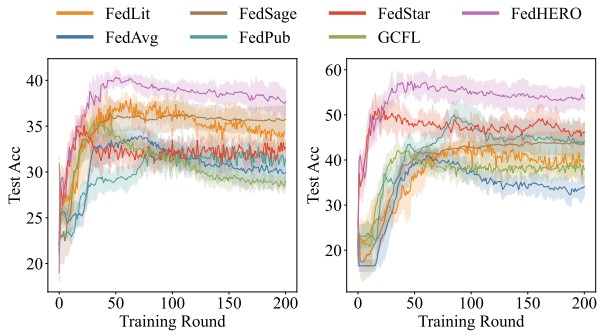

Figure 4: Convergence analysis of FEDHERO and baselines on (left) Squirrel and (right) Chameleon datasets.

Figure 5: (**Left**) Link inference attack (LIA) accuracies on FEDHERO and FGL baseline methods. (**Right**) Performance of FEDHERO and FGL baseline methods on noisy graph datasets.

## 4.7 Empirical Privacy Preservation Evidence in FedHERO

FEDHERO only transmits the components in the $f_{global}$ channel to the server. This subset contains fewer parameters, thus reducing the volume of sensitive information transmitted. We provide empirical privacy evidence by evaluating resistance to link inference attacks (LIA) (Gong & Liu, 2018) that infer the underlying graph structure from the learned node representations. The LIA attack accuracies on two datasets are presented in Figure 5 (left). In general, LIA struggles to accurately predict the graph structure due to the inherent heterophilic characteristics of the graphs. We can observe that FEDHERO offers better privacy protection compared to the FGL baseline approaches by sharing only a portion of the model, and make LIA

even harder to predict. We leave the certified guarantees (e.g., differential privacy guarantees) and other empirical attack evaluations (e.g., membership inference) for future work.

## 4.8 Robustness Study of FedHERO

We assess the robustness of FEDHERO by introducing noise to the graph, flipping each edge with a probability of 0.1. As shown in Figure 5 (right), while baseline FGL methods experience significant performance degradation under such conditions, FEDHERO maintains its effectiveness. This resilience is due to FED-HERO's unique design: unlike other FGL methods that rely heavily on the original graph structure, FED-HERO shares a global structure learner and performs message passing on a generated latent graph. This approach reduces the impact of noise on the original graph's structure, enhancing FEDHERO's adaptability to noisy environments. Additionally, FEDHERO exhibits reduced reliance on the local graph structure, demonstrating its ability to address the performance deterioration brought by varying neighbor distribution patterns and effectively handle heterophilic graphs.

## 5 Related Work

**Graph Structure Learning.** Graph Neural Networks (GNNs) have emerged as a potent tool for analyzing graph-structured data and have found wide applications in various domains (Zhao et al., 2021a; Wu et al., 2019; Kipf & Welling, 2016; Zhu et al., 2019; Liu et al., 2020; Fan et al., 2019; Luan et al., 2022; Yuan et al., 2022; Dwivedi et al., 2023; Shlomi et al., 2020; Wang et al., 2024). In order to overcome the limitations of GNNs in effectively propagating features within observed structures, recent efforts have aimed at simultaneous learning of graph structures alongside the GNN model (Jin et al., 2020; Liu et al., 2022b; Li et al., 2024; Liu et al., 2023b; Wu et al., 2022). At the heart of graph structure learning lies an encoding function responsible for modeling optimal graph structures, often represented by edge weights (Luo et al., 2021; Kreuzer et al., 2021; Sun et al., 2022; Wang et al., 2020; Gasteiger et al., 2019; Yu et al., 2021). These edge weights can be defined through pairwise distances or learnable parameters. For example, GAUGM (Zhao et al., 2021b) calculates edge weights by taking the inner product of node embeddings, introducing no additional parameters. Franceschi et al. (2019) treat each edge as a Bernoulli random variable and optimize graph structures with the GNN. IDGL (Chen et al., 2020), to make full use of information from observed structures for structure learning, employs a multi-head self-attention network to represent pairwise relationships between nodes, while Li et al. (Li et al., 2018a) adopt a metric learning approach based on the Radial Basis Function (RBF) kernel for a similar goal. Moreover, DGM (Kazi et al., 2022) utilizes the Gumbel-Top-k trick to sample edges from a Gaussian distribution and incorporates reinforcement learning, rewarding edges that contribute to correct classifications and penalizing those leading to misclassifications. Additionally, Zhang et al. (2019) present a probabilistic framework that views the input graph as a random sample from a collection modeled by a parametric random graph model. However, while these methods have shown promise, they assume that training and testing nodes originate from the same graph, and they primarily consider a single graph. In contrast, Zhao et al. (2023) address graph structure learning in a cross-graph setting and propose a comprehensive framework for learning a shared structure learner that can generalize to target graphs without necessitating re-training.

**Federated Graph Learning.** Federated Learning (FL) is a paradigm that facilitates model training across decentralized clients while safeguarding privacy (Li et al., 2020; Liu et al., 2024; Gafni et al., 2022; Wang et al., 2023; Ye et al., 2023; Lei et al., 2023). As the field of FL evolves, a new specialized branch has emerged to address the unique challenges of graph data, known as Federated Graph Learning (FGL) (Liu et al., 2022a; Fu et al., 2024b). This branch can be categorized into graph-level and node-level methods Fu et al. (2024a).

Graph-level FGL methods consider scenarios where each client possesses multiple graph data, as seen in domains like molecular structures. Like traditional FL, it mainly focuses on the issue of non-independent and identically distribution (non-IID) issues across clients (Xie et al., 2021; He et al., 2021; Tan et al., 2023; Zheng et al., 2021; Chen et al., 2021a; Fu et al., 2025). On the other hand, node-level FGL assumes that each client holds a single graph, with the objective of addressing node classification tasks. In this variant, each client usually possesses a subgraph of the larger graph, introducing a scenario where there are missing links

between subgraphs. To address this issue, Zhang et al. introduced FEDSAGE (Zhang et al., 2021), which involves collaboratively learning a missing neighbor generator across clients. However, FEDSAGE could raise privacy concerns. In contrast, Baek et al. (2023) utilize random graphs as inputs to compute similarities between clients' GNNs, subsequently employing these similarities for weighted averaging on the server side. Additionally, Xie et al. (2023) focus on detecting latent link-types during FGL and differentiating message-passing through various types of links using multiple convolution channels. In this study, we address a specific challenge in node-level FGL, namely the distinction in node neighbor distributions across clients. Our approach involves sharing a structure learning model among clients, enabling the learning of message-passing patterns across the graphs. By leveraging this information, we aim to enhance the predictive performance of the local models.

## 6 Conclusion

The heterophilic patterns of graphs can result in variations in the distribution of node neighbors among different graphs, leading to performance deterioration for current FGL algorithms. The FEDHERO framework effectively addresses the challenge of varying neighbor distribution patterns, resolving it by sharing structure learners across clients. The structure learner would generate latent graphs with similar tendencies for different clients. Clients' GNNs that capture these tendencies could be aggregated and shared among clients to harness FGL's ability. By decoupling the structural learning model from the local classification model, we effectively capture and make available structural insights for global sharing. Additionally, our proposed framework maintains the capacity for personalization based on local graph structure and node features. Experimental results affirm that FEDHERO consistently surpasses state-of-the-art methods across various datasets.

## Acknowledgments

This work was supported in part by the National Science Foundation (NSF) under grants 2313110, 2143559, IIS-2006844, IIS-2144209, IIS-2223769, CNS-2154962, BCS-2228534, and CMMI-2411248; the Office of Naval Research (ONR) under grant N000142412636; and the Commonwealth Cyber Initiative (CCI) under grant VV-1Q24-011.

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

## A    Experiment Setups

In our experiments, we utilize seven distinct benchmark datasets, the specifics of which are outlined in Table 6. This table provides detailed statistics for each dataset, including the number of nodes, edges, classes, and node feature dimensions. Additionally, we describe the process of partitioning the original graphs into multiple subgraphs to create semi-synthetic federated graph learning datasets.

For graph partitioning, we employ well-established algorithms commonly used in federated graph learning. Specifically, we utilize the METIS algorithm (Karypis & Kumar, 1997) and the Louvain algorithm (Blondel et al., 2008). With METIS, we ensure an even distribution of nodes across subsets, with each subset representing a client's data in a federated learning context. Using Louvain, we follow the methodology of Baek et al. (2023) and (Zhang et al., 2021). We initially partition the graph and select subgraphs containing a minimum of 50 nodes. The smaller subgraphs are then randomly merged into these larger ones to guarantee adequate training data. For example, if Louvain results in 20 subgraphs but only 5 have over 50 nodes, we consolidate the remaining 15 subgraphs into these 5, effectively simulating a federated learning scenario with 5 clients.

We further provide the description of real-world datasets. The syn-cora dataset (Zhu et al., 2020) generates nodes and edges based on the expected homophily ratio within the graph, with feature vectors of nodes in each class derived by sampling from the corresponding class in the Cora dataset (Yang et al., 2016). The ieee-fraud dataset comprises transaction records for products, with edges determined by the proximity of transaction occurrences. The objective is to predict fraudulent transactions (Howard et al., 2019). Meanwhile, the credit dataset has 30,000 nodes, each representing an individual, and their connections are determined by the similarity of their spending and payment patterns (Yeh & Lien, 2009). The objective is to predict whether an individual will default on their credit card payment or not.

Table 6: Dataset statistics

| Dataset | Squirrel | Chameleon | Actor | Flickr | syn-cora | ieee-fraud | credit |
|---------|----------|-----------|-------|--------|----------|------------|--------|
| Nodes | 5,201 | 2,277 | 7,600 | 89,250 | 14,900 | 144,233 | 30,000 |
| Edges | 198,493 | 31,421 | 15,009 | 899,756 | 29,674 | 28,053,717 | 1,436,858 |
| Features | 2,089 | 2,235 | 932 | 500 | 1433 | 371 | 13 |
| Classes | 5 | 5 | 5 | 7 | 5 | 2 | 2 |

We follow the previous work (Mao et al., 2023) to compute the homophily ratios of both the original graphs and the averaged homophily ratios of partitioned graphs. As in Table 7, the partitioned graphs exhibit comparable homophily ratios when compared to the original graphs. It means that the partitioning process does not compromise the heterophilic property of the graphs.

Table 7: Homophily ratio of different partitioning methods.

| Dataset | Squirrel | Chameleon | Actor | Flickr |
|---------|----------|-----------|-------|--------|
| METIS | 0.22±0.02 | 0.26±0.05 | 0.17±0.02 | 0.33±0.04 |
| Louvain | 0.21±0.02 | 0.27±0.06 | 0.19±0.04 | 0.32±0.04 |
| Origin | 0.22 | 0.25 | 0.22 | 0.32 |

## B    Neighbor node distributions of clients in two datasets

In this section, we also give an example in two real-world datasets: Squirrel and Chameleon (Rozemberczki et al., 2021). In these datasets, nodes correspond to web pages, and edges represent hyperlinks between them. To simulate the FL scenario, we apply the community detection algorithm METIS (Karypis & Kumar, 1997) to partition these graphs, and variations in the structures of the resulting subgraphs also arise. As illustrated in Figure 6, we compare the neighbor distributions of two clients in two datasets, respectively. (The neighbor distributions across all clients can refer to Figure 7.) Each bar illustrates, for a specific class of nodes, the

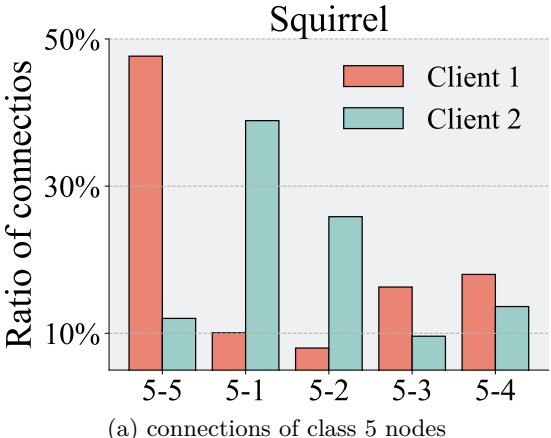 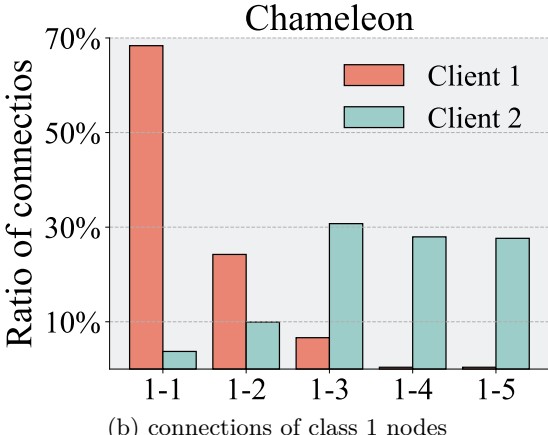

(a) connections of class 5 nodes          (b) connections of class 1 nodes

Figure 6: Neighbor node distributions of two clients in two datasets respectively. Each bar represents the ratio of edges connecting nodes of one class to the total number of edges with nodes of that class as an endpoint. For example, in Figure (a), the orange bar at '5-1' indicates the proportion of edges from class 5 to class 1 nodes relative to total edges originating from class 5 nodes in client 1.

ratio of edges connecting them to other types of nodes of one type to the total number of edges serving this class node as an endpoint. For instance, in Figure 6 (a), the orange bar with the x-axis '5-1' signifies the proportion of edges linking class 5 nodes to class 1 nodes in relation to the overall number of edges originating from class 5 nodes. The results indicate distinct neighbor distributions among different clients. For instance, in the Chameleon dataset, almost 70% of edges originating from class 1 nodes are intra-class connections in client 1, implying class 1 nodes tend to establish hyperlinks with similar web pages. Conversely, less than 10% of the neighbors of class 1 nodes are nodes in the same class in client 2, indicating a preference for establishing hyperlinks with diverse web pages. To safeguard data privacy, the structural details of subgraphs cannot be directly shared between clients. Consequently, this leads to distinct message-passing patterns of information transfer between these subgraphs. Traditional FL methods like FedAvg (McMahan et al., 2017) simply aggregate GNN models, potentially hindering the global model's ability to account for the aggregation discrepancies among various subgraphs effectively.

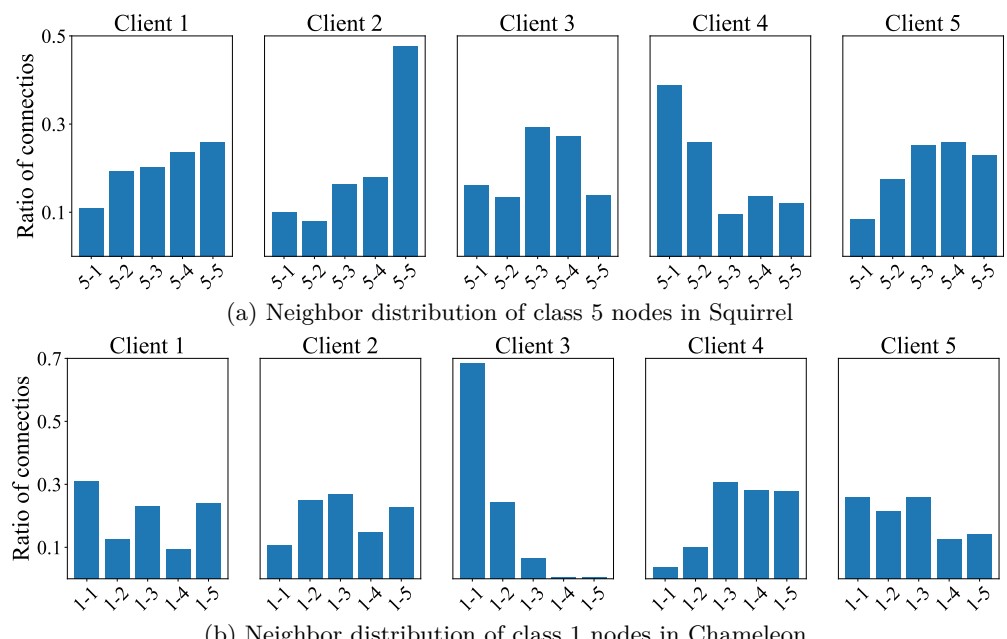

(a) Neighbor distribution of class 5 nodes in Squirrel

(b) Neighbor distribution of class 1 nodes in Chameleon

Figure 7: Neighbor node distributions of clients in two datasets split by METIS.

# C   Case study

In this section, we present a series of case studies designed to further illustrate the practical applications and effectiveness of FEDHERO under diverse conditions. These case studies encompass a range of scenarios, each highlighting different aspects of our approach.

## C.1   Overlapping subgraphs scenario

Despite the commonality of the non-overlapping node assumption, some research investigates scenarios with overlapping nodes. We have followed the data partitioning approach from (Baek et al., 2023) in our experiments, focusing on the Flickr and Actor datasets. The results, as illustrated in Table 8, show FEDHERO consistently outperforming most baselines across these datasets. This indicates its robust capability to manage overlapping scenarios effectively. We theorize that overlapping subgraphs provide client subgraphs with additional connections, hinting at the potential for developing FL algorithms specifically designed to cater to overlapping environments.

Table 8: Results on the overlapping subgraphs scenario.

| Method | LOCAL | FEDAVG | FEDPUB | FEDSTAR | GCFL | FEDLIT | FEDSAGE | FEDHERO |
|---|---|---|---|---|---|---|---|---|
| Flickr | 55.70±0.25 | 55.34±0.69 | 55.50±0.31 | 55.92±0.42 | 56.47±0.21 | 56.64±0.50 | 56.70±0.42 | **56.74±0.30** |
| Actor | 32.46±0.38 | 32.30±1.21 | 28.67±0.89 | 27.98±0.77 | 42.48±1.23 | 42.50±0.66 | 41.57±0.89 | **43.46±1.11** |

## C.2   Latent graph generation

In graph structure learning, there are various methods for generating latent graphs. In our approach, we define $\tilde{a}_{(i,j)}$ as the learned edge weight between nodes $u$ and $v$, and subsequently apply a post-processing step to form a $k$NN graph (where each node is connected to up to $k$ neighbors) as the latent graph. This choice is informed by the differentiability of the top $k$ function, which aids in calculating parameter gradients and updating the model in FEDHERO.

We also examine the influence of another prevalent latent graph generation method (Wu et al., 2020a; Elinas et al., 2020; Zhao et al., 2023). This method treats each edge in the latent graph as a Bernoulli random variable, resulting in edge distributions represented by a product of $N \times N$ independent Bernoulli variables.

Table 9 presents an analysis of FEDHERO's performance when employing the Bernoulli sampling method for latent graph generation during inference. Using $\tilde{a}_{(i,j)}$ as a Bernoulli parameter introduces greater variance, potentially leading to less stable model performance. Despite this, FEDHERO still surpasses most baselines, as evidenced by the comparative results in Table 3. Our experiments also suggest that an asymmetric latent graph may impact model performance. Investigating optimal strategies for utilizing the Bernoulli sampling method in latent graph generation offers a promising direction for future research and development in our work.

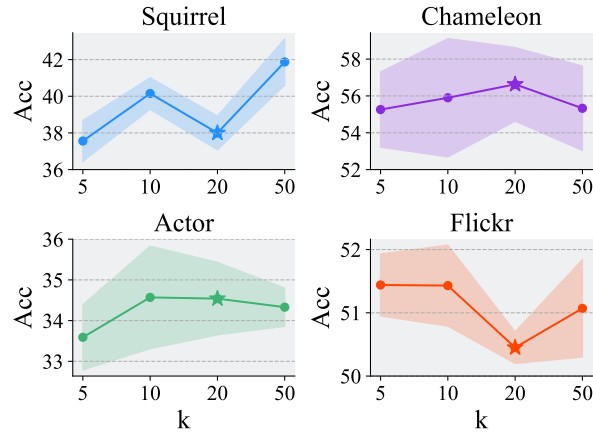

Figure 8: The sensitivity study of hyperparameter $k$. The star point in the figure corresponds to the default result reported in the main text.

Table 9: Results on the different latent graph generation.

| Datasets | Squirrel | | Chameleon | |
|---|---|---|---|---|
| | METIS M=7 | Louvain | METIS M=7 | Louvain |
| Bernoulli | 33.62±1.81 | 36.97±4.43 | 48.82±3.52 | 51.59±2.99 |
| top $k$ | **36.08±0.87** | **38.00±0.92** | **52.65±1.33** | **55.63±2.00** |

To assess the impact of the node degree $k$ in the latent graph, we conducted tests with $k$ set to 5, 10, 20, and 50. Generally, a larger $k$ value results in a denser latent graph. The outcomes of these tests are illustrated in Figure 8. When these findings are combined with the baseline methods' performance from Table 2 in the main paper, we can deduce the following:

- Across varying values of $k$, FEDHERO consistently outperforms all the baseline methods.

- For datasets with a smaller average node degree, such as Actor and Flickr, selecting a relatively smaller $k$ value is advantageous. Conversely, for datasets with a larger average node degree, like Squirrel and Chameleon, opting for a larger $k$ value is preferable.

These insights provide valuable guidance for choosing appropriate $k$ values based on the characteristics of the dataset, thereby optimizing the performance of FEDHERO.

### C.3 Federated graph learning baselines

We provide detailed explanations of the federated graph learning baselines employed in our experiments. These include:

- GCFL (Xie et al., 2021): It incorporates a gradient sequence-based clustering mechanism using dynamic time warping. This mechanism enables multiple data owners with non-IID structured graphs and diverse features to collaboratively train robust graph neural networks.

- FEDSTAR (Tan et al., 2023): It relies on structural embeddings and a feature-structure decoupled graph neural network to address non-IID data challenges through shared structural knowledge. Within the FL framework, it exclusively exchanges the structure encoder. This allows clients to collectively learn global structural knowledge while maintaining personalized learning of node representations based on their features.

- FEDLIT (Xie et al., 2023): It tackles the variability in link characteristics within graphs, where ostensibly uniform links may convey varying degrees of similarity or significance. It dynamically identifies latent link types through an EM-based clustering algorithm during federated learning and adapts message-passing strategies using multiple convolution channels tailored to different link types.

- FEDPUB (Baek et al., 2023): It leverages functional embeddings of the local GNNs with random graphs as inputs to calculate similarities between them. These similarities are then employed for weighted averaging during server-side aggregation. Additionally, the method learns a personalized sparse mask at each client, allowing for the selection and update of only the subset of aggregated parameters relevant to the specific subgraph.

- FEDSAGE (Zhang et al., 2021): It introduces NeighGen, an innovative missing neighbor generator integrated with federated training procedures. This endeavor is geared towards achieving a universal node classification model within a distributed subgraph framework, eliminating the need for direct data sharing.

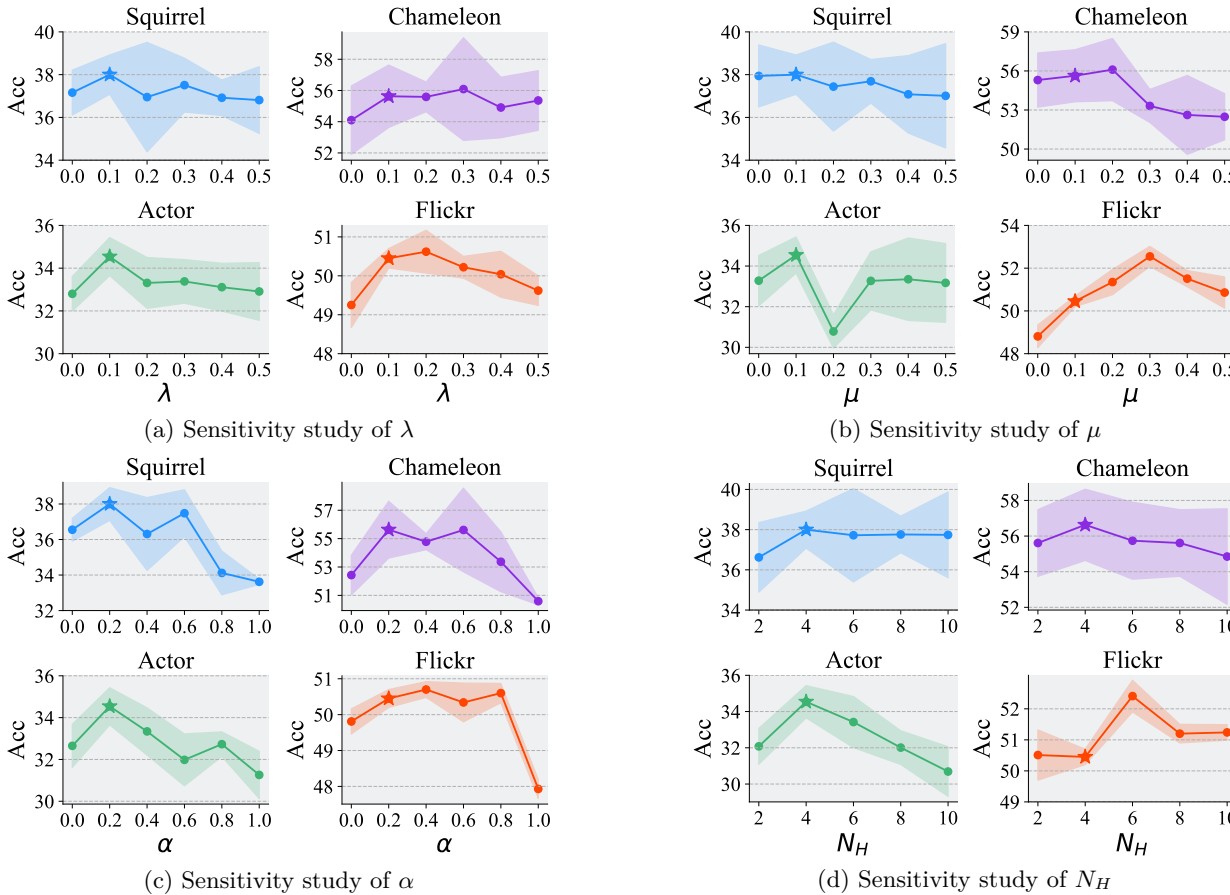

Figure 9: Performance of FEDHERO with different hyperparameter settings. The star point in the figure corresponds to the default result reported in the main text.

# D    Influence of Hyperparameters in FedHERO

## D.1    Hyperparameter Settings.

The architecture of the GNN models consists of a feature projection layer, two graph convolutional layers, and a classifier layer. We employ Adam optimizer (Kingma & Ba, 2014) with a learning rate of 0.005. We apply a consistent hyperparameter setting in our main results. Specifically, we set $\alpha = 0.2$, $\mu = \lambda = 0.1$, $k = 20$, and $N_H = 4$. The training process involves 200 communication rounds, with each local training epoch lasting 1 iteration. The nodes are randomly distributed into five groups. For each experiment, one group is chosen as the test set, while the remaining groups are utilized for training and validation. We perform five experiments on each dataset and report the models' average performance. For the ieee-fraud and credit datasets, we present the mean test Area Under the Curve (AUC), accounting for the imbalanced labels. For the remaining datasets, we use mean test accuracy across clients for evaluation.

In an ideal scenario, a grid search for hyperparameter optimization would be conducted to determine the optimal combination for each set of experiments. It's noteworthy that our experimental results consistently demonstrate the superior performance of FEDHERO compared to all baseline methods, even without hyperparameter tuning. Additionally, we conducted an investigation into the impact of these hyperparameters under the Louvain data partitioning method to provide recommended values for these aggregation and smoothing hyperparameters and their potential impact. The results are illustrated in Figure 9.

| Method | Chameleon | | Squirrel | |
|---|---|---|---|---|
| | Time ↓ (s) | GPU ↓ (MB) | Time ↓ (s) | GPU ↓ (MB) |
| FEDLIT | 35.74 | 899.16 | 69.31 | 3994.69 |
| FEDSTAR | **10.87** | 663.10 | **20.17** | 1020.89 |
| FEDHERO | 11.62 | **601.76** | 20.83 | **948.50** |

Table 10: Average wall-clock training time and peak GPU memory (lower is better) under the Louvain split for 200 rounds on a single RTX A6000.

### D.2  Influence of Smoothing Hyperparameters $\lambda$ and $\mu$

No distinct trend related to the intrinsic nature of the data was observed for $\lambda$ and $\mu$. However, it is noteworthy that removing regularization on structures (i.e., setting $\lambda$ or $\mu$ to 0) resulted in more pronounced performance degradation. In fact, the regularization loss plays a constructive role in offering guidance for structure learning, as highlighted in prior work (Zhao et al., 2023). Conversely, increasing $\lambda$ or $\mu$ to a certain value proved beneficial in enhancing the performance of FEDHERO. However, further increases had a negative impact on the model's focus on the classification task, leading to a decline in performance.

### D.3  Influence of Aggregation Hyperparameters $\alpha$

For the Squirrel, Chameleon, and Actor datasets, optimal results were observed at $\alpha = 0.2$, while $\alpha = 0.4$ exhibited the best performance for the Flickr dataset, as illustrated in Figure 9(c). This variation in optimal $\alpha$ values is attributed to the homophily ratios of the datasets. According to Mao et al. (2023) and Kim & Oh (2022), the homophily ratios for Squirrel, Chameleon, and Actor are 0.22, 0.25, and 0.22, respectively. In contrast, Flickr has a higher homophily ratio of 0.32. Given Flickr's relatively elevated homophily ratio, employing a larger $\alpha$ to increase the proportion of local models proves beneficial, effectively leveraging the structural information within the subgraph. Conversely, for datasets with lower homophily ratios, superior performance is achieved by placing more emphasis on the homophily properties introduced by the global structure learner. In summary, we posit that, for graph data with a larger homophily ratio (indicating lower heterophily), higher values of $\alpha$ can be advantageous.

### D.4  Influence of the Number of Heads $N_H$

Figure 9(d), we investigate FEDHERO's performance under varying values of $N_H$. We observe that increasing $N_H$ up to a certain point enhances FEDHERO's performance. This improvement can be attributed to the aggregation of multi-head results, bolstering the model's ability to capture underlying influences between nodes from diverse perspectives.

However, beyond a certain threshold, further increases in $N_H$ do not yield additional performance improvements. This phenomenon could be attributed to factors such as the saturation of connections between nodes, potential overfitting concerns, and the increased time and computing resources required with larger $N_H$.

## E  Practical Scalability: Runtime and Memory

In this section, we complement the memory and running time analysis (Table 2) with practical metrics-wall-clock training time and peak GPU memory-on Chameleon and Squirrel under the Louvain split for 200 communication rounds, using a single NVIDIA RTX A6000. Each configuration is repeated five times, and we report the average cost.

From Table 10, we can observe that FEDHERO matches FEDSTAR in runtime (within 3–7%) while using less memory on both datasets (9% lower on Chameleon; 7% lower on Squirrel). Compared to FEDLIT , FEDHERO is 67–70% faster and reduces memory by 33% (Chameleon) and 76% (Squirrel). These empirical results align with the analysis in Table 2, indicating that FEDHERO is comparable with FGL baselines and scales well in practice.

## F   Comparative Analysis with FedStar

In this section, we compare FEDHERO with FEDSTAR. Both decouple learning into a feature channel and a structure-focused channel to mitigate feature heterogeneity. FEDHERO, however, targets the harder heterophily regime whereFEDSTAR's static structure embeddings fail and adds (i) a learnable latent-graph generator and (ii) per-layer dual-channel integration, as explained below:

- **Problem setting.** FEDSTAR addresses cross-domain FGL under feature heterogeneity. In contrast, FEDHERO formally studies heterophily in FGL for the first time, where neighbor label distributions can be divergent or contradictory; under strong heterophily, FEDSTAR's assumptions break and its performance degrades.

- **Structure information embedding.** FEDSTAR uses static, hand-crafted structural embeddings (degree, RWSE) without modifying the adjacency. FEDHERO learns a task-aligned latent graph via a multi-head attention metric, dynamically reconstructing edges to suppress misleading cross-class links and enable robust cross-client aggregation.

- **Channel interaction.** FEDSTAR's structure-to-feature flow is essentially one-way. FEDHERO performs per-layer dual-channel integration between global (latent) and local (original) channels, stabilizing training and balancing global alignment with local bias.

- **Empirical evidence.** On strongly heterophilic datasets (homophily $\leq$ 0.25), FEDHERO significantly outperforms FEDSTAR. On syn-cora, where degree distributions are identical across classes Zhu et al. (2020), FEDHERO achieves a $\sim$20% accuracy gain: static degree/RWSE features cannot distinguish nodes with similar degrees but different label contexts, whereas the learned latent graph can.

In summary, while both methods separate global and local channels, FEDHERO's focus on heterophily, dynamic structure learning, and layer-wise dual-channel integration are substantive advances that drive its superior performance on heterophilic graphs in federated settings.

