# OpenReview forum: "FedHERO: A Federated Learning Approach for Node Classification Task on Heterophilic Graphs"
_TMLR — Accepted by TMLR_

### Review · Reviewer_7zGn · 2025-07-15

**Summary Of Contributions:**

The paper proposes FedHERO, a federated graph learning framework for node classification in settings where clients possess heterophilic graphs. Unlike conventional federated graph learning (FGL) approaches that rely on homophily assumptions and suffer performance degradation under varying neighbor-label patterns, FedHERO employs a dual-channel GNN architecture for each client. The global channel processes a latent graph generated by a shared structure learner, while the local channel operates on the client’s original graph. By only aggregating and sharing global-channel weights, FedHERO enables collaborative learning of transferable message-passing patterns across clients while retaining graph-specific information locally. Experimental results show that FedHERO consistently outperforms existing methods across diverse datasets.

**Audience:**

Yes

**Claims And Evidence:**

Yes

**Requested Changes:**

1. The dual-channel “feature + structure” split and the idea of aggregating only the structure branch already appear in FedStar; the authors themselves devote Appendix E to explaining the overlap. The remaining differences (heterophily framing, attention-based metric) feel incremental. As a result, the novelty of FedHero relative to FedStar and other dual-channel FGL frameworks remains unclear. Furthermore, the authors does not articulate a technical difference big enough to justify “first to tackle heterophily in FGL”.

2. Equation (4) defines the latent-edge selector as a binary top-k function and claims this design is “based on the differentiability of the top-k function” ﻿. In reality, top-k is non-differentiable; practical graph-structure-learning papers resort to continuous relaxations. The paper supplies neither a surrogate nor a gradient derivation, creating doubts about whether the reported implementation matches the description.

3. The main text explicitly “assumes no overlapping nodes are shared across data owners” ﻿, but Appendix C evaluates an “overlapping subgraphs scenario” with no algorithmic change except re-running the code ﻿. This raises two issues: the stated assumptions are violated, and it is unclear why FedHERO should work under overlap if its aggregation analysis relies on disjoint subgraphs. A rigorous exposition should either drop the assumption or adapt the method/analysis.

4. Table 2 only lists Big-O complexities and argues FedHERO’s dual-channel cost is “comparable” ﻿, yet no wall-clock time, GPU memory, or transmitted-bytes measurements are reported. For a federated system, such metrics are central; omitting them weakens the argument that the method scales to large client fleets.

5. Symbols such as H, Z, and “UPD/AGG” sometimes change meaning between Sections 3.1 and 3.3. Explicitly list dimensions at first use, and adopt a single bold/italic style for matrices and vectors.

**Strengths And Weaknesses:**

Strengths：

1. Authors performs comprehensive experiments across four real-world graph datasets and three synthetic datasets. The results demonstrate that FedHERO consistently outperforms existing FGL alternatives significantly.
2. FedHERO introduces a dual-channel GNN: a global channel whose parameters are aggregated across clients, and a local channel kept private. By letting the global channel operate on a shared latent graph while the local channel works on the original graph, the design simultaneously extracts universally useful message-passing patterns and preserves client-specific nuances without forcing them into the aggregation.

Weaknesses：

1. The dual-channel “feature + structure” split and the idea of aggregating only the structure branch already appear in FedStar; the authors themselves devote Appendix E to explaining the overlap. The remaining differences (heterophily framing, attention-based metric) feel incremental. As a result, the novelty of FedHero relative to FedStar and other dual-channel FGL frameworks remains unclear. Furthermore, the authors does not articulate a technical difference big enough to justify “first to tackle heterophily in FGL”.

2. Equation (4) defines the latent-edge selector as a binary top-k function and claims this design is “based on the differentiability of the top-k function” ﻿. In reality, top-k is non-differentiable; practical graph-structure-learning papers resort to continuous relaxations. The paper supplies neither a surrogate nor a gradient derivation, creating doubts about whether the reported implementation matches the description.

3. The main text explicitly “assumes no overlapping nodes are shared across data owners” ﻿, but Appendix C evaluates an “overlapping subgraphs scenario” with no algorithmic change except re-running the code ﻿. This raises two issues: the stated assumptions are violated, and it is unclear why FedHERO should work under overlap if its aggregation analysis relies on disjoint subgraphs. A rigorous exposition should either drop the assumption or adapt the method/analysis.

4. Table 2 only lists Big-O complexities and argues FedHERO’s dual-channel cost is “comparable” ﻿, yet no wall-clock time, GPU memory, or transmitted-bytes measurements are reported. For a federated system, such metrics are central; omitting them weakens the argument that the method scales to large client fleets.

5. Symbols such as H, Z, and “UPD/AGG” sometimes change meaning between Sections 3.1 and 3.3. Explicitly list dimensions at first use, and adopt a single bold/italic style for matrices and vectors.

---

> ### Author Response · Authors · 2025-08-14
> **Response from the Authors (1)**
>
> >**W1.&Q1.** The dual-channel “feature + structure” split and the idea of aggregating only the structure branch already appear in FedStar. The remaining differences feel incremental. The novelty of FedHero relative to FedStar and other dual-channel FGL frameworks remains unclear. The authors does not articulate a technical difference big enough to justify “first to tackle heterophily in FGL”.
> >
> **Response**:
>
> Thank you for noting the resemblance between our dual-channel architecture and FedStar’s. While both share the principle of global aggregation over a structure-related branch, FedHERO is the first to explicitly formulate and tackle heterophily in FGL, a setting where FedStar’s static degree/RWSE embeddings are insufficient. Our method introduces a learnable structure learner and layer-wise fusion mechanism that are specifically designed to align structural patterns across heterophilic graphs, as detailed below:
>
> - **Different problem formulation**: FedStar is designed for _cross-domain/dataset_ FGL under **feature heterogeneity**. In contrast, FedHERO is the _first work_ to formally define and study **heterophily in FGL**, where clients’ neighbor distribution patterns can be drastically different or even contradictory. This setting invalidates the assumptions underpinning FedStar, and our experiment results show that FedStar’s performance degrades significantly under high heterophily.
>
> - **Dynamic structure learning vs. static embeddings**: FedStar’s structure encoder is a static, hand-crafted structural embedding (degree-based + RWSE) that **does not alter the original adjacency matrix**. FedHERO introduces a learnable latent-graph generator via a multi-head attention metric function that **dynamically reconstructs edges** from node embeddings. This design is critical in heterophilic graphs: it can break “different-class edges” in the original graph and generate structures that better align with the downstream task, enabling cross-client aggregation even when local neighbor patterns diverge.
>
> - **Layer-wise dual-channel interaction**: While FedStar’s structure channel is used in every layer, it only injects structure information into the feature channel in a **one-way manner**. Instead, **FedHERO fuses the outputs of the global (latent graph) and local (original graph) channels** at every layer in a bi-directional,  enabling both channels to inform each other. This richer interaction is key to stabilizing training and balancing global alignment with local bias in heterophilic graphs.
>
> - **Empirical evidence that the differences matter**: On datasets with strong heterophily (homophily ratio ≤ 0.25), FedHERO significantly outperforms FedStar, showing that our modifications are not cosmetic but necessary for tackling heterophily. We would also like to bring up the **large gap** (20% accuracy gap) on syn-cora, where graphs are random with identical degree distributions across classes [1]. In this setting, **FedStar’s structure embeddings cannot distinguish between nodes with the same degree but very different label contexts, nor between graphs with identical adjacency but different class distributions.** FedHERO’s learnable structure learner bypasses this limitation by reconstructing a latent graph tailored to the prediction task, yielding substantial gains.
>
> [1] Beyond homophily in graph neural networks: Current limitations and effective designs. NIPS, 2020.
>
> ___
>
> >**W2.&Q2.** Equation (4) defines the latent-edge selector as a binary top-k function and claims this design is “based on the differentiability of the top-k function” ﻿. In reality, top-k is non-differentiable; practical graph-structure-learning papers resort to continuous relaxations. The paper supplies neither a surrogate nor a gradient derivation, creating doubts about whether the reported implementation matches the description.
> >
> **Response**:
>
> Thank you for pointing out the confusion around our statement that the design is “based on the differentiability of the top-k function.” We agree that the hard top-k operator is non-differentiable in the strict mathematical sense due to discrete index selection.
>
> After double-checking our implementation, we found that Eq.(4) in the manuscript contains a typo: **the constant 1 should be replaced with $\tilde{a}_{ij}$**. In our actual implementation, we do not directly use the binary top-k values as edge weights. Instead, the top-k result is used as a binary mask to select entries from the attention  matrix $(\tilde{a}_{ij})_{N\times N}$. During backpropagation, topk propagates gradients to the selected k entries by treating the selection mask as fixed during the backward pass. This allows sparse but direct updates to the parameters corresponding to the active edges.
>
> We will revise the manuscript to (1) correct the expression in Eq.(4), and (2) replace the phrase “based on the differentiability of the top-k function” with a precise explanation of this masking mechanism to avoid ambiguity.

---

> > ### Author Response · Authors · 2025-08-14
> > **Response from the Authors (2)**
> >
> > >**W3.&Q3.** The main text explicitly “assumes no overlapping nodes are shared across data owners” ﻿, but Appendix C evaluates an “overlapping subgraphs scenario” with no algorithmic change except re-running the code ﻿. This raises two issues: the stated assumptions are violated, and it is unclear why FedHERO should work under overlap if its aggregation analysis relies on disjoint subgraphs. A rigorous exposition should either drop the assumption or adapt the method/analysis.
> > >
> > **Response**:
> >
> > Thank you for the careful reading. We would like to clarify the following points:
> > - The phrase “assume no overlapping nodes are shared across data owners” appears **_only_** in Sec. 4.1 (Experimental Settings) as an evaluation protocol choice for comparability with prior FGL baselines (e.g., FedPub and FedSage). **Our method does not require non-overlapping clients.**
> >
> > - The algorithm aggregates only global-channel parameters via a FedAvg-style weighted average. No component relies on unique nodes among subgraphs, cross-client neighborhoods, or graph stitching; raw data never leaves a client. Thus, overlap does not affect correctness of the method or its analysis.
> >
> > - Appendix C reports an overlapping-subgraphs study, also considered in FedPub, to demonstrate extensibility. The identical code path and training loop are used, and performance remains strong, no special handling needed.
> >
> > Non-overlap thus serves _only_ as a widely considered scenatio for FGL setting, not as a methodological constraint.
> >
> > ___
> >
> > >**W4.&Q4.** Table 2 only lists Big-O complexities and argues FedHERO’s dual-channel cost is “comparable” ﻿, yet no wall-clock time, GPU memory, or transmitted-bytes measurements are reported. For a federated system, such metrics are central; omitting them weakens the argument that the method scales to large client fleets.
> > >
> > **Response**:
> >
> > We appreciate your suggestion to complement the Big-O complexity analysis with practical metrics. In addition to the theoretical analysis in Table 2, we have now measured wall-clock training time and peak GPU memory consumption for FGL methods on the Chameleon and Squirrel datasets (Louvain split, 200 communication rounds, repeated 5 times, single NVIDIA RTX A6000 GPU):
> >
> > The results show that FedHERO uses GPU memory comparable to FedStar, while being faster and  memory-efficient than FedLit on larger dataset. These empirical findings are consistent with our Big-O analysis, reinforcing that FedHERO is scalable to large client fleets in practice as well as in theory.
> >
> >
> > | Method | Dataset | Running Time(s) | GPU Memory (MB) |
> > | - | - | - | - |
> > | FedLit  | Chameleon  | 35.74 | 899.16 |
> > | FedStar    | Chameleon     | 10.87   | 663.10 |
> > | FedHERO     | Chameleon     | 11.62   | 601.76|
> > | FedLit  | Squirrel  | 69.31 | 3994.69 |
> > | FedStar    | Squirrel     | 20.17  | 1020.89 |
> > | FedHERO     | Squirrel     | 20.83  | 948.5 |
> >
> >
> > ___
> >
> > >**W5.&Q5.** Symbols such as H, Z, and “UPD/AGG” sometimes change meaning between Sections 3.1 and 3.3. Explicitly list dimensions at first use, and adopt a single bold/italic style for matrices and vectors.
> > >
> > **Response**:
> >
> > Thank you for pointing out the notational inconsistencies between Sections 3.1 and 3.3. In the revised version, we will: (i) unify the meaning of symbols such as H, Z, and UPD/AGG throughout the paper; (ii) explicitly state the dimensions of each symbol at first use; and (iii) adopt a consistent style to avoid confusion.

---

> > > ### Comment · Reviewer_7zGn · 2025-09-09
> > >
> > > Thank you for the author's reply, and the author has partially addressed my questions.

---

> > > > ### Author Response · Authors · 2025-09-11
> > > > **Response from the Authors**
> > > >
> > > > Thank you for your follow-up. We're glad to hear that our rebuttal was helpful. If there are any remaining concerns or points you'd like us to clarify further, we’d be happy to do so.

---

### Review · Reviewer_hcAc · 2025-08-04

**Summary Of Contributions:**

This paper addresses challenge in Federated Graph Learning (FGL) where the performance of Graph Neural Network (GNN) models deteriorates when clients' local graphs exhibit varying degrees of heterophily. The authors propose FedHERO to handle this issue. FedHERO proposes a dual-channel GNN architecture for each client. A global channel operates on a latent graph generated by a shared structure learner, aiming to capture universally applicable patterns that can be aggregated across clients. A separate local channel operates on the original client graph to learn personalized, client-specific patterns. The authors conduct extensive experiments on several semi-synthetic and real-world datasets, demonstrating that FedHERO generally outperforms existing FGL baselines, particularly on graphs with high heterophily.

**Audience:**

Yes

**Claims And Evidence:**

Yes

**Requested Changes:**

I would recommend acceptance of this paper without necessary changes.

**Strengths And Weaknesses:**

**Strengths**
- Relevant and Well-Defined Problem: The paper successfully identifies and formulates a practical and important problem in FGL. The assumption that client graphs are homophilic is often violated in real-world scenarios, and addressing the challenge of varying neighbor distributions across clients is a valuable contribution to the field.
- Novel Dual-Channel Architecture: The proposed FedHERO architecture is an intuitive and clever solution. The decoupling of the model into a shared global channel that learns from a generated, more uniform latent graph and a private local channel that retains client-specific information is an effective design for balancing generalization and personalization.
- Thorough Experimental Evaluation: The empirical validation is comprehensive. The authors use a variety of datasets with different characteristics, employ multiple graph partitioning methods, and compare FedHERO against a strong and relevant set of recent FGL baselines. The inclusion of extensive ablation studies on sharing mechanisms, structure learning methods, and hyperparameters further strengthens the paper's claims.


**Weaknesses**
- Incremental Novelty Over Baselines: The proposed method has considerable conceptual overlap with the FedStar baseline, particularly the idea of decoupling the GNN and sharing only a structure-focused component.  Although the paper includes a direct comparison, the novelty of FedHERO can be seen as more incremental than transformative, which slightly diminishes the paper's impact.

---

> ### Author Response · Authors · 2025-08-14
> **Response from the Authors**
>
> >**W1.** Incremental Novelty Over Baselines: The proposed method has considerable conceptual overlap with the FedStar baseline, particularly the idea of decoupling the GNN and sharing only a structure-focused component. Although the paper includes a direct comparison, the novelty of FedHERO can be seen as more incremental than transformative, which slightly diminishes the paper's impact.
> >
> **Response**:
>
> We appreciate your comment regarding the conceptual overlap with FedStar. Indeed, both frameworks decouple the model and share a structure-focused component. This is by design, as structure information can help align clients under feature heterogeneity. However, FedHERO extends this idea to a different and more challenging setting—heterophily in FGL, where FedStar’s static structure embeddings fail, and addresses it with a learnable latent-graph generator and per-layer dual-channel integration, as explained below:
>
> - **Different problem formulation**: FedStar is designed for _cross-domain/dataset_ FGL under **feature heterogeneity**. In contrast, FedHERO is the _first work_ to formally define and study **heterophily in FGL**, where clients’ neighbor distribution patterns can be drastically different or even contradictory. This setting invalidates the assumptions underpinning FedStar, and our experiment results show that FedStar’s performance degrades significantly under high heterophily.
>
> - **Dynamic structure learning vs. static embeddings**: FedStar’s structure encoder is a static, hand-crafted structural embedding (degree-based + RWSE) that **does not alter the original adjacency matrix**. FedHERO introduces a learnable latent-graph generator via a multi-head attention metric function that **dynamically reconstructs edges** from node embeddings. This design is critical in heterophilic graphs: it can break “different-class edges” in the original graph and generate structures that better align with the downstream task, enabling cross-client aggregation even when local neighbor patterns diverge.
>
> - **Layer-wise dual-channel interaction**: While FedStar’s structure channel is used in every layer, it only injects structure information into the feature channel in a **one-way manner**. Instead, **FedHERO fuses the outputs of the global (latent graph) and local (original graph) channels** at every layer in a bi-directional,  enabling both channels to inform each other. This richer interaction is key to stabilizing training and balancing global alignment with local bias in heterophilic graphs.
>
> - **Empirical evidence that the differences matter**: On datasets with strong heterophily (homophily ratio ≤ 0.25), FedHERO significantly outperforms FedStar, showing that our modifications are not cosmetic but necessary for tackling heterophily. We would also like to bring up the **large gap** (20% accuracy gap) on syn-cora, where graphs are random with identical degree distributions across classes [1]. In this setting, **FedStar’s structure embeddings cannot distinguish between nodes with the same degree but very different label contexts, nor between graphs with identical adjacency but different class distributions.** FedHERO’s learnable structure learner bypasses this limitation by reconstructing a latent graph tailored to the prediction task, yielding substantial gains.
>
> In summary, while we acknowledge conceptual parallels in separating “global” and “local” channels, FedHERO’s problem framing (heterophily in FGL), dynamic structure learning mechanism, and layer-wise channel fusion are novel and technically substantial. These innovations are key to FedHERO’s consistently superior performance and make it the first framework tailored to effectively handle heterophilic graphs in federated settings.
>
> [1] Beyond homophily in graph neural networks: Current limitations and effective designs. NIPS, 2020.

---

### Review · Reviewer_mXFF · 2025-08-10

**Summary Of Contributions:**

This paper proposes FedHERO, a federated graph learning framework designed to handle heterophilic graphs, where connected nodes often have different labels. FedHERO employs a dual-channel GNN: the global channel uses a shared structure learner to generate latent graphs that capture universal structural patterns across clients, while the local channel operates on each client’s original graph to retain personalized information. Experiments are conducted on both semi-synthetic and real-world datasets.

**Audience:**

Yes

**Broader Impact Concerns:**

This work has the potential for broader impact, which is not currently discussed in the paper.

**Claims And Evidence:**

Yes

**Requested Changes:**

1. Please provide a clearer and more compelling explanation of the fundamental novelty in your proposed dual-channel design compared to FedStar (Tan et al., 2023) and other personalized FL frameworks. In particular, clarify why the incorporation of a structure learner represents a substantive innovation rather than an incremental adaptation.


2. The proposed structure learner appears to be a direct adaptation of IDGL (Chen et al., 2020) to a federated setting. We recommend that you either (a) introduce non-trivial theoretical or algorithmic improvements, or (b) provide a rigorous analysis explaining why this adaptation is both challenging and significant in the context of federated learning.


3. Please provide stronger empirical evidence that the heterophily setting in your experiments faithfully reflects real-world federated heterophilic graph scenarios. Consider incorporating additional real-world heterogeneous-client datasets beyond ieee-fraud and credit, ideally with varying and quantified heterophily levels. If synthetic partitions are used, please include a detailed justification of how these partitions reflect realistic data distributions. For reference, here are examples of benchmark datasets in federated learning: [A], [B] (other benchmarks may also be suitable for the revision).
[A] https://arxiv.org/abs/2007.13518
[B] https://arxiv.org/abs/2104.07145


4. Please avoid using the same constant learning rate and other fixed hyperparameters across all experiments without justification. We suggest including more detailed hyperparameter tuning methods, such as grid search. At a minimum, include sensitivity analyses to demonstrate how different hyperparameter settings influence model performance, particularly in heterogeneous and noisy environments.


5. Since privacy protection is a key claimed advantage, please support this claim with formal methods (e.g., differential privacy guarantees) or empirical attack evaluations (e.g., membership inference, attribute inference). Clearly quantify the privacy benefits of sharing only partial model components. If such analyses cannot be provided, the privacy-related claims should be substantially downplayed.

**Strengths And Weaknesses:**

Strengths:

1. The proposed dual-channel design balances global knowledge sharing with local personalization.
2. Empirical results show consistent gains in accuracy and demonstrate robustness to noise.


Weaknesses:

1. The idea of splitting a model into a global channel and a local channel is already present in FedStar (Tan et al., 2023) and other personalized FL frameworks. The authors do not convincingly argue what is fundamentally new here, apart from incorporating a structure learner.


2. The proposed structure learner (multi-head attention similarity + kNN) appears to be a straightforward adaptation of IDGL (Chen et al., 2020) to a federated context. The paper offers limited substantial theoretical or algorithmic innovation in graph structure learning.

3. The experiments are not sufficiently convincing. While they motivate heterophily as a problem, the specific setting (different levels of heterophily across clients) is not strongly supported by real federated heterophilic graph datasets—most are synthetic partitions of existing graphs. Specifically, the “heterophilic” setting is largely created via METIS/Louvain partitioning of homophilic graphs, which may not faithfully represent truly heterogeneous-client scenarios in real FGL. Moreover, only a few small “real” datasets are used (e.g., ieee-fraud, credit). These are not compelling for GNN-specific claims.

4. The hyperparameter settings are overly simplistic. For example, the authors use the same constant learning rate across different settings.

5. The privacy argument is neither quantitative nor convincing. The authors merely claim that the proposed method could provide stricter privacy protection, but this is not supported by formal privacy analysis. Typically, to claim improved privacy, one should use empirical privacy attack evaluations (e.g., membership inference attacks) or formal theoretical tools such as differential privacy. Even though sharing only part of the model may yield some privacy benefits, a quantitative assessment of the privacy of the proposed algorithm is essential for a research paper. Without such analysis, privacy should not be presented as a key selling point.

---

> ### Author Response · Authors · 2025-08-14
> **Response from the Authors (1)**
>
> >**W1.&Q1.** Please provide a clearer and more compelling explanation of the fundamental novelty in FedHERO compared to FedStar (Tan et al., 2023) and other personalized FL frameworks. In particular, clarify why the incorporation of a structure learner represents a substantive innovation rather than an incremental adaptation.
> >
> **Response**:
>
> We appreciate your observation and agree that both FedHERO and FedStar employ a dual-channel design with a globally shared structure-focused component. This similarity is intentional, both approaches address the challenge that feature heterogeneity can harm cross-client knowledge transfer, with structure information serving as an alignment signal. However, we would like to clarify **FedHERO is not merely FedStar plus a structure learner**, as detailed below:
>
> - **Different problem formulation**: FedStar is designed for _cross-domain/dataset_ FGL under **feature heterogeneity**. In contrast, FedHERO is the _first work_ to formally define and study **heterophily in FGL**, where clients’ neighbor distribution patterns can be drastically different or even contradictory. This setting invalidates the assumptions underpinning FedStar, and our experiment results show that FedStar’s performance degrades significantly under high heterophily.
>
> - **Dynamic structure learning vs. static embeddings**: FedStar’s structure encoder is a static, hand-crafted structural embedding (degree-based + RWSE) that **does not alter the original adjacency matrix**. FedHERO introduces a learnable latent-graph generator via a multi-head attention metric function that **dynamically reconstructs edges** from node embeddings. This design is critical in heterophilic graphs: it can break “different-class edges” in the original graph and generate structures that better align with the downstream task, enabling cross-client aggregation even when local neighbor patterns diverge.
>
> - **Layer-wise dual-channel interaction**: While FedStar’s structure channel is used in every layer, it only injects structure information into the feature channel in a **one-way manner**. Instead, **FedHERO fuses the outputs of the global (latent graph) and local (original graph) channels** at every layer in a bi-directional,  enabling both channels to inform each other. This richer interaction is key to stabilizing training and balancing global alignment with local bias in heterophilic graphs.
>
> - **Empirical evidence that the differences matter**: On datasets with strong heterophily (homophily ratio ≤ 0.25), FedHERO significantly outperforms FedStar, showing that our modifications are not cosmetic but necessary for tackling heterophily. We would also like to bring up the **large gap** (20% accuracy gap) on syn-cora, where graphs are random with identical degree distributions across classes [1]. In this setting, **FedStar’s structure embeddings cannot distinguish between nodes with the same degree but very different label contexts, nor between graphs with identical adjacency but different class distributions.** FedHERO’s learnable structure learner bypasses this limitation by reconstructing a latent graph tailored to the prediction task, yielding substantial gains.
>
> In summary, while we acknowledge conceptual parallels in separating “global” and “local” channels, FedHERO’s problem framing (heterophily in FGL), dynamic structure learning mechanism, and layer-wise channel fusion are novel and technically substantial. These innovations are key to FedHERO’s consistently superior performance and make it the first framework tailored to effectively handle heterophilic graphs in federated settings.
>
> [1] Beyond homophily in graph neural networks: Current limitations and effective designs. NIPS, 2020.

---

> > ### Author Response · Authors · 2025-08-14
> > **Response from the Authors (2)**
> >
> > >**W2.&Q2.** The proposed structure learner appears to be a direct adaptation of IDGL (Chen et al., 2020) to a federated setting. We recommend that you either (a) introduce non-trivial theoretical or algorithmic improvements, or (b) provide a rigorous analysis explaining why this adaptation is both challenging and significant in the context of federated learning.
> > >
> > **Response**:
> >
> > Thank you for your insightful comments and the suggestion to clarify the relationship between our structure learner and IDGL (Chen et al., 2020). We would like to emphasize the following points:
> >
> > - **Similarity + kNN is a common paradigm, not unique to IDGL.** While our structure learner adopts a “multi-head attention similarity + kNN” approach, this design pattern is **widely used** in the literature for graph structure refinement or construction [1], not exclusive to IDGL.
> >
> > - **Different objectives from IDGL.** The central purpose of our structure learner is **not** iterative structure optimization for improving a single-task GNN, as in IDGL. Instead, our goal is to encode structural knowledge into a transferable structure learner model that can be shared across clients in a FGL environment without exposing raw graph data. This enables clients with heterogeneous (particularly heterophilic) graphs to collaborate by learning a latent structural representation that captures cross-client commonalities.
> >
> > - **Framework novelty beyond the structure learner.** While the structure learner is a key component, **the novelty of FedHERO does not hinge solely on this module**. FedHERO introduces a dual-channel architecture, a cross-client shared structure learning mechanism, and a training paradigm that jointly addresses heterophily-induced knowledge inconsistency and privacy-preserving collaboration.
> >
> > In summary, although our similarity+kNN component may appear superficially similar to parts of IDGL, its role, integration, and application context in FedHERO are fundamentally different, addressing challenges that are absent in the centralized IDGL setting
> >
> > [1] Wentao Zhao, et al. Graphglow: Universal and generalizable structure learning for graph neural networks. KDD 2023.
> >
> > >**W3.&Q3.** Please provide stronger empirical evidence that the heterophily setting in your experiments faithfully reflects real-world federated heterophilic graph scenarios. Consider incorporating additional real-world heterogeneous-client datasets beyond ieee-fraud and credit, ideally with varying and quantified heterophily levels. If synthetic partitions are used, please include a detailed justification of how these partitions reflect realistic data distributions. For reference, here are examples of benchmark datasets in federated learning: [A], [B] (other benchmarks may also be suitable for the revision). [A] https://arxiv.org/abs/2007.13518 [B] https://arxiv.org/abs/2104.07145
> > >
> > **Response**:
> >
> > We appreciate your concern regarding the realism of our heterophily setting and the representativeness of our datasets. However, we would like to clarify and provide further evidence:
> >
> > - **We already adopted heterophilic graph datasets in our experiments.** As demonstrated in Table 7 in Appendix A, all four datasets have homophily ratio less than 0.32, and they are widely treated as heterophilic graphs [1]. In Appendix A and B, we also quantitatively show that after partitioning, clients **indeed** hold different and measurable heterophily ratios, supporting the intended heterogeneous-client setting.
> >
> > - **The adopted synthetic partitions are common and accepted in FGL.** We acknowledge that our setting uses synthetic client partitions. This is consistent with widespread practice in FGL (e.g., FedSage, FedPub), especially when large-scale, labeled, naturally partitioned heterophilic datasets are scarce.
> >
> >
> > - **Limitations of suggested benchmarks [A] and [B].** We thank you for pointing to [A] and [B]. However:
> >
> >     1. [A] focuses on standard FL datasets (e.g., image classification) and **does not include any FGL datasets.**
> >
> >     2. **[B] only includes homophilic graphs** (e.g., Cora (homo ratio=0.81), PubMed (homo ratio=0.80) [1]) and still **depends on manual partitioning** (e.g., LDA-based splits) to simulate realistic FGL setting. It does not contain naturally heterogeneous graph datasets.
> >
> >
> > - **On real-world datasets.** We agree with you that incorporating large-scale, naturally partitioned real-world heterophilic graph datasets would further strengthen empirical validity. However, the creation and public availability of such datasets remain an open challenge for the community and are beyond the scope of this work.
> >
> > In summary, our experimental design that partitions heterophilic graphs with METIS/Louvain preserves cross-client heterogeneity and aligns with prevailing FGL practices. We have clarified this in the main text and highlighted the quantitative evidence in Appendix A and B.
> >
> >
> > [1] Yao Ma, et al. Is homophily a necessity for graph neural networks? ICLR 2022.

---

> > > ### Author Response · Authors · 2025-08-14
> > > **Response from the Authors (3)**
> > >
> > > >**W4.&Q4.**  Please avoid using the same constant learning rate and other fixed hyperparameters across all experiments without justification. We suggest including more detailed hyperparameter tuning methods, such as grid search. At a minimum, include sensitivity analyses to demonstrate how different hyperparameter settings influence model performance, particularly in heterogeneous and noisy environments.
> > > >
> > > **Response**:
> > >
> > > Thank you for the suggestion regarding hyperparameter tuning. The experimental setup intentionally uses a uniform set of hyperparameters across all datasets to avoid dataset-specific fine-tuning advantages and to provide a fair, controlled comparison with baselines. Even under this fixed configuration, FedHERO **consistently** achieves superior performance relative to all baselines in both semi-synthetic and real-world settings, demonstrating the method’s robustness without the need for dataset-specific tuning.
> > >
> > > However, we would also like to highlight that **a comprehensive hyperparameter analysis is already included in Appendix D**. This analysis evaluates performance sensitivity over a wide range of values. We will clarify in the revised paper that the constant hyperparameter choice in the main experiments was made to ensure comparability with prior work, and that Appendix D already provides the requested sensitivity analyses.
> > >
> > > ___
> > >
> > > >**W5.&Q5.**  Since privacy protection is a key claimed advantage, please support this claim with formal methods (e.g., differential privacy guarantees) or empirical attack evaluations (e.g., membership inference, attribute inference). Clearly quantify the privacy benefits of sharing only partial model components. If such analyses cannot be provided, the privacy-related claims should be substantially downplayed.
> > > >
> > > **Response**:
> > >
> > > Thank you for the comment regarding the need for quantitative privacy assessment. However, we would like to note that we already include an empirical privacy evaluation using the Link Inference Attack (LIA) in Sec. 4.7, which aims to recover the underlying graph structure from the learned node representations. The LIA attack accuracies on two datasets are shown in Fig. 5 in our manuscript.
> > >
> > > The results show that LIA struggles to accurately reconstruct the graph structure in our heterophilic setting, and that **FedHERO consistently achieves lower LIA attack accuracy compared to FGL baselines.** This improvement stems from sharing only the global-channel parameters rather than the full model, which reduces direct exposure of client-specific structural information.
> > >
> > > We will make this clearer in the revision by explicitly framing our privacy argument around the LIA results in Sec. 4.7, and by downplaying broader privacy claims beyond this empirical evidence. While we acknowledge that formal guarantees such as differential privacy are not included, the LIA results quantitatively support that FedHERO provides stronger resistance to structure inference attacks compared to existing FGL approaches. We also recognize that incorporating other privacy attack evaluations like membership inference and formal theoretical tools such as differential privacy is essential for a more comprehensive privacy analysis, and we plan to explore these directions as important future work.

---

### Comment · Reviewer_7zGn · 2025-07-07

The paper proposes FedHERO, a federated graph learning framework for node classification in settings where clients possess heterophilic graphs. Unlike conventional federated graph learning (FGL) approaches that rely on homophily assumptions and suffer performance degradation under varying neighbor-label patterns, FedHERO employs a dual-channel GNN architecture for each client. The global channel processes a latent graph generated by a shared structure learner, while the local channel operates on the client’s original graph. By only aggregating and sharing global-channel weights, FedHERO enables collaborative learning of transferable message-passing patterns across clients while retaining graph-specific information locally. Experimental results show that FedHERO consistently outperforms existing methods across diverse datasets.

 Strengths：
1. Authors performs comprehensive experiments across four real-world graph datasets and three synthetic datasets. The results demonstrate that FedHERO consistently outperforms existing FGL alternatives significantly.
2. FedHERO introduces a dual-channel GNN: a global channel whose parameters are aggregated across clients, and a local channel kept private. By letting the global channel operate on a shared latent graph while the local channel works on the original graph, the design simultaneously  extracts universally useful message-passing patterns and preserves client-specific nuances without forcing them into the aggregation.

Weaknesses：
1. The dual-channel “feature + structure” split and the idea of aggregating only the structure branch already appear in FedStar; the authors themselves devote Appendix E to explaining the overlap. The remaining differences (heterophily framing, attention-based metric) feel incremental. As a result, the novelty of FedHero relative to FedStar and other dual-channel FGL frameworks remains unclear. Furthermore, the authors does not articulate a technical difference big enough to justify “first to tackle heterophily in FGL”.
2. Equation (4) defines the latent-edge selector as a binary top-k function and claims this design is “based on the differentiability of the top-k function” ﻿. In reality, top-k is non-differentiable; practical graph-structure-learning papers resort to continuous relaxations. The paper supplies neither a surrogate nor a gradient derivation, creating doubts about whether the reported implementation matches the description.
3. The main text explicitly “assumes no overlapping nodes are shared across data owners” ﻿, but Appendix C evaluates an “overlapping subgraphs scenario” with no algorithmic change except re-running the code ﻿. This raises two issues: the stated assumptions are violated, and it is unclear why FedHERO should work under overlap if its aggregation analysis relies on disjoint subgraphs. A rigorous exposition should either drop the assumption or adapt the method/analysis.
4. Table 2 only lists Big-O complexities and argues FedHERO’s dual-channel cost is “comparable” ﻿, yet no wall-clock time, GPU memory, or transmitted-bytes measurements are reported. For a federated system, such metrics are central; omitting them weakens the argument that the method scales to large client fleets.

5. Symbols such as H, Z, and “UPD/AGG” sometimes change meaning between Sections 3.1 and 3.3. Explicitly list dimensions at first use, and adopt a single bold/italic style for matrices and vectors.

---

### Decision · Action_Editor_kv9r · 2025-11-07

**Recommendation:** Accept with minor revision

**Additional Comments:**

Most reviewer concerns were satisfactorily clarified in the rebuttal, but several improvements should be explicitly incorporated into the main text. The authors should integrate the key explanations provided in their response, particularly
- the distinction between FedHERO and FedStar, emphasizing the learnable latent-graph generator and layer-wise bi-directional fusion;
- the correction and clarification of the top-k masking mechanism;
- the rationale that the non-overlapping clients assumption serves only as an evaluation protocol;
- the added runtime and memory measurements that support scalability claims.

In addition, the manuscript should explicitly highlight the scope and limitation of its privacy evidence, noting that the Link Inference Attack provides empirical but not formal privacy guarantees. Incorporating these clarifications into the main text will ensure the final version is fully self-contained, transparent, and consistent with the authors’ rebuttal explanations.

**Audience:**

Yes

**Audience Explanation:**

The topic is clearly relevant to TMLR’s readership. The reviewers agree that addressing heterophily in FGL is timely and practically meaningful, especially given the privacy constraints in cross-client collaboration. The proposed design should be of the interests to TMLR’s audience in distributed and privacy-preserving ML, as well as graph learning.

**Claims And Evidence:**

Yes

**Claims Explanation:**

The paper is technically sound and well supported by experiments.

The authors’ rebuttal clarified key issues: FedHERO differs from FedStar by defining heterophily in federated graph learning and introducing a learnable latent-graph generator with layer-wise bidirectional fusion. They corrected the top-k differentiability claim, explained that the non-overlapping assumption was only an evaluation setting, and added runtime and memory evidence confirming scalability. These clarifications strengthen the work’s validity.

However, the contribution remains somewhat incremental, as the dual-channel design parallels prior frameworks and privacy claims lack formal guarantees. Overall, the revised submission presents convincing empirical evidence and clear methodology, though with moderate conceptual novelty.